# Enhancing Tail Performance in Extreme Classifiers by Label Variance Reduction

**Anirudh Buvanesh**\*, **Rahul Chand**\*, **Jatin Prakash, Bhawna Paliwal, Mudit Dhawan**
**Neelabh Madan, Deepesh Hada, Vidit Jain, Sonu Mehta, Yashoteja Prabhu**
**Manish Gupta, Ramachandran Ramjee, Manik Varma**

Microsoft
{t-abuvanesh, t-rahulchand, t-japrakash, bhawna, t-mdhawan
t-nmadan, deepeshhada, jainvidit, sonu.mehta, yprabhu
gmanish, ramjee, manik}@microsoft.com

## Abstract

Extreme Classification (XC) architectures, which utilize a massive One-vs-All (OvA) classifier layer at the output, have demonstrated remarkable performance on problems with large label sets. Nonetheless, these architectures falter on tail labels with few representative samples. This phenomenon has been attributed to factors such as classifier over-fitting and missing label bias, and solutions involving regularization and loss re-calibration have been developed. This paper explores the impact of label variance - a previously unexamined factor - on the tail performance in extreme classifiers. It also develops a method to systematically reduce label variance in XC by transferring the knowledge from a specialized tail-robust teacher model to the OvA classifiers. For this purpose, it proposes a principled knowledge distillation framework, LEVER, which enhances the tail performance in extreme classifiers with formal guarantees on generalization. Comprehensive experiments are conducted on a diverse set of XC datasets, demonstrating that LEVER can enhance tail performance by around $5\%$ and $6\%$ points in PSP and coverage metrics, respectively, when integrated with leading extreme classifiers. Moreover, it establishes a new state-of-the-art when added to the top-performing Renée classifier. Extensive ablations and analyses substantiate the efficacy of our design choices. Another significant contribution is the release of two new XC datasets that are different from and more challenging than the available benchmark datasets, thereby encouraging more rigorous algorithmic evaluation in the future. Code for LEVER is available at: aka.ms/lever.

## 1 Introduction

Extreme Classification (XC) addresses tasks where a data point is mapped to the most relevant *subset* of labels from a large label space. Deep architectures that comprise a neural network encoder followed by a massive One-vs-All (OvA) classification layer at the output have become the de-facto standard for contemporary XC algorithms and have demonstrated remarkable results on several large-scale applications (Agrawal et al., 2013; Yadav et al., 2021; Chang et al., 2020; Beygelzimer et al., 2009; Babbar & Schölkopf, 2017). Despite this progress, such over-parameterized OvA classification layers has also been known to overfit and underperform on labels with limited representative samples, also known as the *tail* labels (Wei et al., 2021). As a result, bulk of such tail labels, which often provide niche and highly informative results for a test sample (Jain et al., 2016), are incorrectly classified thus diminishing their aggregate utility for a practical application.

The challenge of enhancing the tail performance of extreme OvA classifiers has been the focus of some recent studies. These investigations have identified multiple factors that contribute to the hardness of tail labels and proposed solutions to alleviate them. Some works have addressed the

---

\*Primary Authors. Correspondence to: Anirudh Buvanesh <anirudhb1102@gmail.com>, Yashoteja Prabhu <yprabhu@microsoft.com>

concern of overfitting to data-scarce tail labels by constraining the capacity of tail classifiers through regularization tricks (Guo et al., 2019). A separate line of work has studied the effects of false negatives, also known as missing labels, on the tail performance and proposed to appropriately amend the classifier training loss through propensity-scoring techniques (Qaraei et al., 2021).

This paper brings to light another important yet previously unexamined factor behind the under-performance of tail OvA classifiers, namely *label variance* (Sec. 3). Typically, the ground truth of an XC dataset is constructed by approximating a complex label distribution that arises in a source application with a discrete sample of labels. For example, in a recommendation task, the ground truth is defined as the set of items clicked by each user within a specified time period. However, in general, the ground truth can vary from one data sampling period to another, as a user's interests can fluctuate with time. Similarly, in expert annotation based data, employing fewer experts to reduce annotation costs can introduce variance in the ground truth (*aka.* label variance) owing to inter-annotator disagreements. Large label variance is particularly harmful for the tail classifiers' performance as they have to rely on sparse ground truth and the approximation errors can have a drastically magnified effects with low sample counts.

In a recent work, Menon et al. (2021a) studied the problem of label variance in the context of multi-class classification and retrieval, and further note that a teacher-to-student knowledge distillation strategy can be used to improve the generalization performance of the student model. This paper borrows the basic ideas from Menon et al. (2021a) and extends them to the more challenging Extreme Classification setting through several key innovations. First, it theoretically formalizes the performance degradation in OvA classifiers owing to label variance, specifically quantifying the magnified effect on the tail classifiers. Second, whereas Menon et al. (2021a) assumes the pre-existence of a teacher, this paper learns its own Siamese-style teacher model that is optimized for tail performance, and further develops a principled knowledge distillation strategy to effectively teach the downstream OvA classifiers. The resulting approach, LEVER, is demonstrated to improve tail classifiers' performance by around $5\%$ and $6\%$ points in terms of PSP and coverage metrics, which also advances the state-of-the-art in XC.

Another independent contribution of this paper is the public release of two new datasets for algorithmic benchmarking in XC. Traditionally, performance in XC is mostly assessed on the public datasets available from (Bhatia et al., 2016). These datasets appear to share a common property that the data points associated with a label are fairly similar to each other in their semantic intents, making these datasets less challenging to learn. In contrast, the real-world applications of XC can be more diverse in their properties and complexity. To encourage more rigorous algorithmic evaluation, the new datasets are constructed with the property that a label can be associated with data points of vastly different intents. These datasets, termed as *multi-intent* datasets, are inspired by real applications, are more challenging, and can unlock exciting research problems in the future.

This paper makes the following key contributions: *1*. Identifies the problem of label variance which adversely affects the performance of tail classifiers in XC. *2*. Proposes a principled LEVER approach to mitigate the label variance effects on tail classifiers in XC (Sec. 3.2). *3*. Develops an effective Siamese-style model as a tail teacher with LEVER (Sec. 3.3). *4*. Conducts extensive experimentation using multiple state-of-the-art baselines and diverse benchmarks to demonstrate the utility and generality of the proposed approach (Sec. 5). *5*. Releases two new multi-intent datasets for robust experimentation in XC (Sec. 4).

## 2 RELATED WORK

### 2.1 EXTREME CLASSIFICATION

Recent advancements in XC have leveraged deep network-based representations like LSTM (You et al., 2018), Transformer (Zhang et al., 2021; Jiang et al., 2021) or customized architectures (Dahiya et al., 2021b) to generate rich semantic representations of inputs. These are then assigned to appropriate labels via an OvA classifier layer. To facilitate efficient learning with large label sets, techniques such as multi-staged encoder refinement (Dahiya et al., 2021a; Zhang et al., 2021; Jiang et al., 2021), hierarchical label search, and hard-negative sampling (Dahiya et al., 2023a; 2021b; Zhang et al., 2021; Jiang et al., 2021; Mittal et al., 2021a) have been introduced. Furthermore, simultaneous training of the deep encoder and OvA classifiers has been demonstrated to boost per-

formance in leading XC approaches like DEXA (Dahiya et al., 2023b), ELIAS (Gupta et al., 2022), CascadeXML (Kharbanda et al., 2022) and Renée (Jain et al., 2023). However, despite these advancements, many of these approaches share a common limitation: a decline in performance for tail labels, which is the primary focus of this paper.

## 2.2 ENHANCING TAIL PERFORMANCE IN XC

Extreme classifiers have been observed to under-perform on tail labels with limited representative samples. This phenomenon has been attributed to various factors, and several approaches have been proposed to address them.

**Over-fitting of OvA Classifiers**: OvA classifiers, which employ a distinct classifier for each label, are massively parameterized in scenarios with large label sets. Consequently, they are susceptible to overfitting on tail labels with scarce representative samples. In response, various classifier regularization techniques have been introduced. For instance, ProXML (Babbar & Schölkopf, 2019) employs an L1-regularizer, and GLaS (Guo et al., 2019) uses a label-decorrelation based regularizer.

**Bias due to Missing Labels**: In XC datasets, which are often too large for exhaustive labeling, missing or false negative labels are a frequent issue. These missing labels introduce systematic biases into the ground truth and are known to significantly impact tail labels. Strategies to address tail labels typically involve estimating the missing propensities for labels first and then recalibrating the loss through simple weighting (Jain et al., 2016; Wei et al., 2021; Wydmuch et al., 2021; Schultheis et al., 2022). The phenomenon of missing label bias is distinct from that of label variance.

**Data Scarcity in Tail Labels**: XC datasets contain tail labels with a limited number of positive data samples. To mitigate this scarcity, data augmentation techniques like TAUG (Wei et al., 2021) and Gandalf (Kharbanda et al., 2024) have been proposed. However, these methods lack formal guarantees and do not perform consistently across different datasets as shown in this paper (Table 2). Another line of work leverages label-side features to improve the tail label prediction performance (Xiong et al., 2020; Dahiya et al., 2021a; 2023a; Jain et al., 2023). Approaches like NGAME (Dahiya et al., 2023a) share information between semantically similar labels by placing them close to each other in a dense embedding space using a Siamese encoder. However, these methods primarily focus on enhancing encoder robustness and do not explicitly address the quality of subsequent OvA classifiers. Our proposed model shares similarities with these approaches through its use of a Siamese teacher but distinguishes itself by learning a specialized teacher model suitable for distillation and developing a principled approach to improve tail OvA classifiers.

In addition to these known issues, this paper introduces label variance as an additional, but important, consideration pertaining to tail performance in XC. A closely related work is the study around uncertainty quantification in extreme classification (Jiang et al., 2023) because variance can intrinsically be viewed as an uncertainty measurement. But in this work, we attempt to mitigate variance rather than just estimate it. It is important to differentiate the label variance discussed here from the variance described in (Babbar & Schölkopf, 2019). The latter addresses variance from the perspective of lack of commonality between the features of train and test instances. In contrast, our focus on label variance pertains to inaccuracies in the ground truth relevance scores.

## 3 LEVER: LABEL VARIANCE REDUCTION IN EXTREME CLASSIFICATION

Label variance is a measure of approximation errors introduced in the ground truth of a dataset due to the discrete data sampling process. These errors can negatively impact the performance of trained classifiers, particularly those on the tail. This section introduces LEVER, a principled approach based on knowledge distillation designed to alleviate label variance and enhance the generalization capabilities of One-vs-All (OvA) classifiers. An effective teacher model for distillation based on a Siamese-style encoder is also proposed.

### 3.1 PRELIMINARIES

Extreme Classification (XC) maps a data point space $\mathcal{X}$ onto a label space represented as $\mathcal{Y} = \{0, 1\}^L$, where $L$ is the number of labels, potentially reaching into the millions. A deep extreme classification architecture typically includes a deep encoder $\mathcal{E}_\theta$ which generates a semantically rich

representation $\mathcal{E}_\theta(\mathbf{x})$ for any given input data point $\mathbf{x} \in \mathcal{X}$. This is followed by a One-vs-All classifier layer $\{\mathbf{w}_l\}_{l=1}^{L}$ which sorts the labels based on $\mathbf{w}_l^\top \mathcal{E}_\theta(\mathbf{x})$ scores and predicts the highest scoring labels as the most relevant ones for $\mathbf{x}$.

Different strategies have been employed for training such a deep architecture including stagewise training where encoder and classifiers are optimized in two successive stages, and end-to-end training where both are optimized jointly. For this paper, we assume a stagewise training schedule. Furthermore, the focus will be primarily on the second stage of OvA classifier training during which encoder is assumed to be already trained and held fixed. As a result, each OvA classifier is trained independently of others which also simplifies the theoretical analysis. For brevity, we drop the encoder symbol $\mathcal{E}_\theta$ and directly use $\mathbf{x}$ to refer to a data point's embedding from the encoder over which OvA classifiers are applied.

For a data point $\mathbf{x}$, let $\mathbb{P}(Y(\mathbf{x}) = \mathbf{y}|\mathbf{x})) \;\; \forall \mathbf{y} \in \{0, 1\}^L$ represent the true and complete distribution of label relevance which accurately captures the stochasticities inherent in the user preferences or annotator judgments. Note that this distribution sums up to 1 over all label subsets. Unfortunately, the full relevance distribution is seldom available and is instead approximated with a discrete sample of labels $\mathbf{y} \sim \mathbb{P}(Y(\mathbf{x}) = y|\mathbf{x})$. The approximation error due to this sampling is captured by the following expression for label variance:

$$\mathbb{V}_{\mathbf{y}|\mathbf{x}}[\mathbf{y}] = \mathbb{E}_{\mathbf{y}|\mathbf{x}}[\mathbf{y} - \mathbb{E}[\mathbf{y}]]^2$$
$$\mathbb{V}_{y_l|\mathbf{x}}[y_l] = \mathbb{E}_{y_l|\mathbf{x}}[y_l - \mathbb{E}[y_l]]^2 = \mathbb{P}(y_l = 1|\mathbf{x})(1 - \mathbb{P}(y_l = 1|\mathbf{x})) \tag{1}$$

The second expression denotes the variance in the marginal relevance of a label $l$ to point $\mathbf{x}$, a term that is particularly useful in analyzing One-vs-All classifiers. A larger variance indicates that the imprecision in a sampled label is more.

To train the classifier for label $l$, we first construct a training set denoted as $\mathcal{D} = \{\mathbf{x}_i, y_{il}\}_{i=1}^{N}$ and solve a binary classification problem with $y_{il}$ as the target label for $\mathbf{x}_i$. For simplicity, we present the analysis for a single classifier, with the understanding that the same holds for all classifiers. To avoid confusion, we omit subscript $l$ where it is not necessary. The binary classification objective minimizes the following empirical risk of classification:

$$\hat{\mathbf{R}} = \min_{\mathbf{w}} \frac{1}{N} \sum_{i=1}^{N} \mathcal{L}(y_i, \mathbf{w}^\top \mathbf{x}_i)$$
$$\text{with,} \;\; \mathcal{L}(y, \mathbf{w}^\top \mathbf{x}) = Cyf(1, \mathbf{w}^\top \mathbf{x}) + (1-y)f(0, \mathbf{w}^\top \mathbf{x}) \tag{2}$$

Here, $f$ represents a convex classification surrogate such as hinge loss or logistic loss (Qaraei et al., 2021). Using a weight factor $C > 1$ is standard practice in imbalanced classification to appropriately balance the relative importance of positive and negative samples for a label. This is particularly important for a tail label with a few positives, denoted by number $S$ where:

$$\mathbb{E}_{\mathbf{x}}[p_x] \approx \frac{S}{N} \ll 1 \;\; \text{where,} \;\; p_x = \mathbb{P}(y = 1|\mathbf{x}) \tag{3}$$

Following the standard practice (Kakade et al., 2008), we assume that the norms of the weight vector $\mathbf{w}$ and the input vector $\mathbf{x}$ are bounded by $\|\mathbf{w}\| \leq W$ and $\|\mathbf{x}\| \leq B$ respectively. Additionally, we assume that the function $f$ exhibits Lipschitz continuity with a Lipschitz constant $L$.

The generalization performance of a trained classifier $\mathbf{w}$ is evaluated by its true population risk. A lower value of this risk indicates superior predictive capability:

$$\mathbf{R} = \mathbb{E}_{\mathbf{x},y}[\mathcal{L}(y, \mathbf{w}^\top \mathbf{x})] \tag{4}$$

## 3.2 LEVER FRAMEWORK

The deviation between empirical and true risks formally measures a classifier's generalization gap, with smaller values indicating better test-time generalization. Following (Maurer & Pontil, 2009), we express the generalization gap in terms of data-dependent bounds based on label variance. Applying Bennett's inequality, as suggested in the reference, with simplifications relevant to the problem at hand, provides us with the following result. Note that all the proofs are available from the supplementary Sec. A.

**Theorem 1.** *Let $\mathcal{M}_N$ be the uniform covering number (Menon et al., 2021a) corresponding to the classification loss $\mathcal{L}$. Then, given the definitions established earlier, For any $\delta \in (0,1)$, with probability at least $1 - \delta$ over sampling the data points $\{\mathbf{x}\}_{i=1}^N$,*

$$\mathbf{R} \leq \hat{\mathbf{R}} + \mathcal{O}\Big( \sqrt{\mathbb{V}_{\mathbf{x}}[\mathcal{L}(p_x, \mathbf{w}^\top \mathbf{x})] + \mathbb{E}_{\mathbf{x}}[\mathbb{V}_{y|\mathbf{x}}[y|\mathbf{x}]](CLWB)^2}\sqrt{\frac{\log(\mathcal{M}_N/\delta)}{N}} + \frac{\log(\mathcal{M}_N/\delta)}{N} \Big) \tag{5}$$

*where, $\mathbb{V}_{\mathbf{x}}\mathcal{L}(p_x, \mathbf{w}^\top \mathbf{x})$ and $\mathbb{V}_y[y|\mathbf{x}]$ are the variances in the loss function contributed by $\mathbf{x}$, and conditional variance of $y$ respectively.*

**Lemma 1.** *Assuming the loss weighting factor $C$ defined in Eq. 2 as $C = \frac{N}{S}$, where $S$ is the threshold defined in Eq. 3 and $N$ is the number of training points, the variance term $\mathcal{V} = \mathbb{E}_{\mathbf{x}}[\mathbb{V}_y[y|\mathbf{x}]](CLWB)^2$ in Theorem 1 is bounded by $\frac{N(LWB)^2}{S}$.*

Theorem 1 establishes a strong dependence between the classifier performance and the variance in labels $\mathbb{V}_{y|\mathbf{x}}[y|\mathbf{x}]$ with larger values of the latter degrading the effectiveness of the trained classifiers. Furthermore, Lemma 1 shows that a smaller positive sample count $S$ can amplify the adverse effect of label variance which makes the tail classifiers more prone to label variance-related degradation. Now, if we have access to precise estimates of marginal relevance, denoted by $p_x = \mathbb{E}[y|\mathbf{x}]$, we can replace $y$ with $p_x$, effectively reducing the label variance term to 0. This forms the intuition behind LEVER which employs an additional teacher network to provide accurate estimates of $p_x$.

In practice, however, obtaining a perfect teacher is infeasible both due to modeling and computational hardness issues. As a result, the ability to robustly leverage a partially biased teacher to improve the target student model is essential for the practical utility of LEVER. To enable this, we propose the following variant of LEVER where an imperfect teacher's relevance estimates are used for regularizing the original loss with discrete labels:

$$\min_{\mathbf{w}} \frac{\lambda}{N} \sum_{i=1}^N \mathcal{L}(y_i, \mathbf{w}^\top \mathbf{x}_i) + \frac{1-\lambda}{N} \sum_{i=1}^N \mathcal{L}(\hat{p}_i, \mathbf{w}^\top \mathbf{x}_i) \tag{6}$$

where $\hat{p}_i$ are the relevance estimates outputted by the teacher model, and $\lambda$ is a regularization hyperparameter. The above formulation aims to trade off variance errors due to $y_i$ with the bias errors due to $\hat{p}_i$ to attain the lowest overall generalization error. The following theorem shows that, for an appropriate choice of $\lambda$, the risk of the resulting classifier is lower than when trained on either $y_i$ or $\hat{p}_i$ alone:

**Theorem 2.** *Let $\mathbf{R}, \hat{\mathbf{R}}$ be the population risk and empirical risk for a binary classification loss $\mathcal{L}$. Let $\mathcal{M}_N$ be the uniform covering number (Menon et al., 2021a) corresponding to $\mathcal{L}$. Also, let the teacher be imperfect with maximum possible error in relevance estimates bounded by $E = \|p_{\mathbf{x}} - \hat{p}_{\mathbf{x}}\|_\infty$. Then, by solving the regularized optimization problem $\hat{\mathbf{R}}_s = \min_{\mathbf{w}} \frac{\lambda}{N} \sum_{i=1}^N \mathcal{L}(y_i, \mathbf{w}^\top \mathbf{x}_i) + \frac{1-\lambda}{N} \sum_{i=1}^N \mathcal{L}(\hat{p}_i, \mathbf{w}^\top \mathbf{x}_i)$ and setting $\lambda$ to minimize population risk- for any $\delta \in (0,1)$, the following inequality holds with probability at least $1 - \delta$ over sampling the data points $\{\mathbf{x}\}_{i=1}^N$ under the assumption of a reasonably small teacher error $(E)$:*

$$\lambda = \frac{c}{b}\sqrt{\frac{a}{b^2 - c^2}} \quad ; \quad \mathbf{R} \leq \hat{\mathbf{R}}_s + \sqrt{a - a\frac{c^2}{b^2}} + c + \frac{\log(\mathcal{M}_N/\delta)}{N} \tag{7}$$

$$where, \quad a = V_x \frac{\log(\mathcal{M}_N/\delta)}{N} \quad ; \quad b = \sqrt{S\log(\mathcal{M}_N/\delta)}\frac{CLWB}{N} \quad ; \quad c = ECLWB \tag{8}$$

Note that when $c = 0$, $\lambda = 0$ which is equivalent to training on pure teacher estimates. Also, when $0 < c \leq b$, $\sqrt{a - a\frac{c^2}{b^2}} + c \leq \min\{\sqrt{a + b^2}, \sqrt{a} + c\}$. In other words, the bound over population risk is tighter than when $\lambda = 0$ or $\lambda = 1$. Therefore, trading off the teacher's bias with label variance by setting an appropriate $0 < \lambda < 1$ can lead to better generalization than pure training with either original ground truth or biased teacher estimates as label targets.

### 3.3 A Siamese-Style Teacher for LEVER

Recent studies have shown that Siamese Networks, when used as input encoders, exhibit strong performance on tail labels (Dahiya et al., 2021a; 2023a; Jain et al., 2023). This success can be

attributed to the ability of Siamese encoders to leverage label correlations by utilizing label-side features. These features, often presented as descriptive text or structured graphs over labels, are commonly found in XC applications. In fact, most recent XC datasets have started to incorporate them (Bhatia et al., 2016). Consequently, this allows for the sharing of information between semantically similar labels, effectively addressing the problem of data scarcity in tail labels. It is important to note, however, that a standalone Siamese model is insufficient as it tends to under-fit data-rich head labels, thereby compromising overall prediction quality. This paper, therefore, proposes the use of Siamese Networks as teachers within the LEVER framework to enhance the tail performance of one-vs-all classifiers. By employing LEVER, we can improve the tail performance of one-vs-all classifiers without compromising their already excellent head accuracies.

A Siamese encoder, $\mathcal{E}_\theta$, is trained to map the features of data points, denoted as $\{\mathbf{x}_i\}_{i=1}^N$, and label features, represented as $\{\mathbf{z}_l\}_{l=1}^L$, into a common embedding space. The objective of this mapping is to ensure that labels relevant to a given data point are positioned closer in the embedding space, while those that are irrelevant are distanced. Typically, this is achieved by minimizing a triplet loss $[\mathbf{z}_l^\top \mathbf{x}_k - \mathbf{z}_l^\top \mathbf{x}_i + \Delta]_+$, where $k$ and $l$ are a negative and a positive samples, respectively, for label $l$ and $\Delta$ is a margin enforced for better generalization (Dahiya et al., 2021a; 2023a). However, the triplet-loss is not probabilistically calibrated and does not provide reliable marginal relevance targets for training a student. To address this, we leverage a logistic-loss based objective that is found to be well-calibrated:

$$\min_\theta \sum_{l \in L} \sum_{\substack{k \in X_- \\ i \in X_+}} \log(1 + e^{\mathbf{z}_l^\top \mathbf{x}_k - \mathbf{z}_l^\top \mathbf{x}_i + \Delta}) \tag{9}$$

The following theorem demonstrates the calibration property of Eq. 9 assuming that the loss can be fully minimized, i.e., loss between each positive-negative pair is minimized.

**Theorem 3.** *Consider a label $\mathbf{z}$, and a pair of data points $\mathbf{x}_a, \mathbf{x}_b$. Let $p_a, p_b$ be the probabilities that the label is relevant to points $a, b$ respectively. Then, assuming that Eq. 9 is fully minimized, the expected loss in Eq. 9 is minimized for $p_a = 1/(1 + e^{-(\mathbf{z}^\top \mathbf{x}_a + c)}), p_b = 1/(1 + e^{-(\mathbf{z}^\top \mathbf{x}_b + c)})$.*

The above result shows a direct connection between the Siamese model's scores and relevance probabilities, which can be exploited as teacher targets. The parameter $c$ is a hyper-parameter, and it is fitted by cross-validation. While the above strategy provides well-calibrated scores, we empirically observe that simple score mapping strategies, such as $p_a = \frac{\cos \text{Sim}(z, x_a) + 1}{2}$, where $\cos$ Sim represents the cosine similarity, also work equally well.

To make training tractable, we follow the negative mining strategy used in NGAME (Dahiya et al., 2023a). Motivated by recent works that under-sample (or oversample) model inputs (Menon et al., 2021b) to address dataset imbalance, we modify NGAME's point-wise sampling strategy to a label-wise approach, in which mini-batches are made from labels rather than points. This adjustment leads to the up-sampling of tail labels, thereby increasing their importance during training. Empirically, we find that a teacher trained via this strategy exhibits better tail performance. Subsequently, the one-vs-all (OvA) classifier distilled from this teacher outperforms the OvA classifier distilled from the Siamese teacher trained with point-wise sampling, both in precision (+0.37% on average in P@1) and in PSP (+1.7% on average in PSP@1), as detailed in Table 12 in the appendix.

## 4    CONTRIBUTED DATASETS

**Motivation** Performance evaluation of XC algorithms has largely relied on public benchmark datasets available from (Bhatia et al., 2016). In these datasets, the data points associated with a label tend to be fairly similar to each other in their semantic intents. We refer to these as *single-intent* datasets. For example, in LF-AmazonTitles-131K, the label "clothing for men" might be associated with "formal shirts for men" or "casual shirts for men". In contrast, several real-world XC applications belong to a *multi-intent* setting where the label can be associated with data points of vastly different intents. For instance, in query auto-completion (Yadav et al., 2021) where the prefix of a search query needs to be mapped to its completing suffixes, a suffix "..book" might start with either "face.." or "note.." as prefix thus leading to completely different final queries. Such *multi-intent* datasets can be challenging for XC but are under-represented among existing benchmarks. Additionally, the datasets we release exhibit significant imbalances compared to existing benchmarks, with

$7 - 11\%$ of the labels accounting for $80\%$ of the positive instances (refer Table 3). This imbalance poses multiple challenges. First, methods like GLaS (Guo et al., 2019) and Gandalf (Kharbanda et al., 2024), which depend on label correlations for regularization or data augmentation, struggle due to the sparse correlations among tail labels when imbalance is high (refer Table 2). Second, classifier-based methods may achieve high precision by focusing on the head labels, but this results in poor performance on tail metrics such as coverage (refer Table 2). We believe that the contributed datasets will promote further study into developing methods that are robust across various dataset settings.

**Contributed datasets:** Two new datasets, LF-AOL-270K and LF-WikiHierarchy-1M are curated. LF-AOL-270K involves the query auto-completion task of matching a query prefix with completing suffixes. It is curated from publically available AOL search logs (Pass et al., 2006). LF-WikiHierarchy-1M involves the taxonomy completion task (Benaouicha et al., 2016) of matching a Wikipedia category to its parent categories (Zesch & Gurevych, 2007). This dataset is motivated by the real-world application of query-to-ad keyword matching where a keyword can subsume the intent of its query thus giving rise to hierarchical association structures. Complete dataset creation details and dataset statistics are provided in appendix Sec. B.3.

## 5 EXPERIMENTS AND RESULTS

**Datasets**: LEVER was evaluated on a diverse set of datasets, encompassing both full-text and short-text feature scenarios, as well as novel multi-intent datasets. Specifically, we utilized three full-text datasets (LF-Amazon-131K, LF-Wikipedia-500K, LF-WikiSeeAlso-320K), two short-text datasets (LF-AmazonTitles-131K, LF-AmazonTitles-1.3M), and two new multi-intent datasets (LF-WikiHierarchy-1M and LF-AOL-270K). For detailed dataset statistics, please refer to Table 3 in the appendix. Additionally, we evaluate LEVER on a large proprietary query-to-keyword matching dataset with 20M labels (refer Sec. B.2 in the appendix for more details).

**Evaluation Metrics**: To assess the test-time performance, standard evaluation metrics were used, namely precision@k (P@$k$, $k$=1, 3, and 5) and its propensity-weighted variant PSP@$k$ (with $k$=1, 3, and 5). Detailed definitions for these metrics can be found in (Bhatia et al., 2016). Additionally, following the recommendations in (Schultheis et al., 2022), we also included coverage@k (C@$k$) as an important metric to evaluate the tail performance.

**Baselines** We applied LEVER to improve multiple strong OvA-based baselines, including CascadeXML (Kharbanda et al., 2022), ELIAS (Gupta et al., 2022), and Renée (Jain et al., 2023), for demonstrating its effectiveness and generality. We also compared LEVER to other competing tail-enhancement techniques including regularization-based methods such as GLaS (Guo et al., 2019) and $L2$-regularization, data augmentation methods like TAUG (Wei et al., 2021) and Gandalf (Kharbanda et al., 2024), and propensity weighting approaches such as Re-rank (Wei et al., 2021). For comprehensive details on model hyper-parameters, please refer to Sec. D in the appendix.

**LEVER Implementation Details** As discussed in Sec. 3, LEVER uses a Siamese teacher to obtain relevance estimates, $\hat{p}$. Using the relevance estimates an augmented dataset $\mathcal{D}_{aug}$ is created by adding each label as a document, resulting in a dataset comprising $N + L$ documents and $L$ labels. Document and label embeddings from the Siamese teacher are then used to add $\tau_l$ nearest labels, and $\tau_d$ nearest documents for a particular label. In total, $\tau = \tau_l + \tau_d$ elements are added for each label. Empirically, we find that not adding documents ($\tau_d = 0$) leads to performance similar to that of adding documents in most cases. For more details on LEVER's hyper-parameters refer Sec. D.7.1 in the appendix.

**Performance on SOTA OvA methods** Table 1 demonstrates LEVER's effectiveness when applied to leading classifier-based XC methods, including CascadeXML, ELIAS, and Renée. LEVER consistently improves P@1 and PSP@1 on average by 2% and 5%, respectively, across all base models and datasets. When applied to Renée, LEVER achieves new state-of-the-art, increasing PSP@1 by up to 5% while maintaining comparable precision. Notably, LEVER proves highly effective on smaller datasets (LF-AmazonTitles-131K, LF-Amazon-131K), highlighting its importance when data is limited. Table 13 in appendix further illustrates LEVER's gains on a proprietary dataset containing 20M labels. Larger improvements in ELIAS and CascadeXML are attributed to these models not explicitly utilizing label features during training or initialization. In contrast, Renée, which uses

Table 1: LEVER can be applied to improve any OvA-based approach. When used with leading OvA approaches LEVER consistently boosts tail performance across all benchmarks, increasing PSP on average by $5.3\%$ while maintaining comparable precision ($1.4\%$ gain on average). Coverage metrics (reported in Table 5 in the appendix) show similar trends with an average gain of $6.5\%$.

| Model | LF-AmazonTitles-131K | | | | | | LF-Amazon-131K | | | | | |
|---|---|---|---|---|---|---|---|---|---|---|---|---|
| | P@1 | P@3 | P@5 | PSP@1 | PSP@3 | PSP@5 | P@1 | P@3 | P@5 | PSP@1 | PSP@3 | PSP@5 |
| ELIAS | 37.28 | 25.18 | 18.14 | 28.95 | 34.45 | 39.08 | 43.03 | 29.27 | 21.20 | 33.49 | 40.80 | 46.76 |
| ELIAS + LEVER | **42.86** | **28.37** | **20.16** | **36.30** | **41.05** | **45.43** | **47.38** | **32.24** | **23.22** | **38.97** | **46.74** | **52.79** |
| CascadeXML | 36.28 | 24.88 | 18.18 | 26.50 | 33.21 | 38.81 | 43.76 | 29.75 | 21.58 | 34.05 | 41.69 | 47.96 |
| CascadeXML + LEVER | **43.58** | **28.79** | **20.63** | **36.24** | **41.83** | **46.95** | **48.24** | **32.82** | **23.73** | **39.09** | **47.55** | **54.18** |
| Renée | 46.05 | 30.81 | **22.04** | 38.47 | 44.87 | **50.33** | 48.05 | 32.33 | 23.26 | 39.32 | 47.10 | 53.51 |
| Renée + LEVER | **46.44** | **30.83** | 21.92 | **39.70** | **45.44** | 50.31 | **49.19** | **33.30** | **24.04** | **40.64** | **48.48** | **54.87** |

| | LF-Wikipedia-500K | | | | | | LF-AmazonTitles-1.3M | | | | | |
|---|---|---|---|---|---|---|---|---|---|---|---|---|
| | P@1 | P@3 | P@5 | PSP@1 | PSP@3 | PSP@5 | P@1 | P@3 | P@5 | PSP@1 | PSP@3 | PSP@5 |
| ELIAS | 81.94 | 62.71 | 48.75 | 33.58 | 43.92 | 48.67 | 47.48 | 42.21 | 38.60 | 18.79 | 23.20 | 26.06 |
| ELIAS + LEVER | **82.44** | **63.88** | **50.03** | **36.94** | **49.28** | **55.03** | **48.91** | **43.17** | **39.28** | **23.68** | **27.43** | **29.72** |
| CascadeXML | 77.00 | 58.30 | 45.10 | 31.25 | 39.35 | 43.29 | 47.14 | 41.43 | 37.73 | 15.92 | 20.23 | 23.16 |
| CascadeXML + LEVER | **80.10** | **60.41** | **46.44** | **36.79** | **46.65** | **50.99** | **47.98** | **42.02** | **38.12** | **20.06** | **24.51** | **27.28** |
| Renée | 84.95 | 66.25 | 51.68 | 37.10 | 50.27 | 55.68 | **56.10** | **49.91** | **45.32** | 28.56 | 33.38 | 36.14 |
| Renée + LEVER | **85.02** | **66.37** | **51.98** | **42.93** | **55.00** | **60.29** | 56.01 | 49.43 | 44.85 | **33.55** | **36.82** | **38.81** |

| | LF-AOL-270K | | | | | | LF-WikiHierarchy-1M | | | | | |
|---|---|---|---|---|---|---|---|---|---|---|---|---|
| | P@1 | P@3 | P@5 | PSP@1 | PSP@3 | PSP@5 | P@1 | P@3 | P@5 | PSP@1 | PSP@3 | PSP@5 |
| ELIAS | 40.83 | 22.33 | 14.91 | 13.29 | 21.46 | 25.22 | **95.27** | **94.25** | **92.45** | 17.15 | 24.41 | 30.01 |
| ELIAS + LEVER | **40.85** | **22.83** | **15.57** | **13.68** | **24.30** | **30.43** | 94.02 | 91.97 | 89.50 | **28.27** | **36.80** | **42.13** |
| CascadeXML | **41.20** | **22.12** | 14.82 | **12.58** | 19.53 | 23.19 | **94.88** | **93.69** | **91.79** | 16.03 | 22.87 | 28.17 |
| CascadeXML + LEVER | 39.41 | 21.78 | **14.99** | 11.96 | **21.30** | **27.59** | 94.77 | 93.54 | 91.56 | **20.14** | **27.49** | **33.01** |
| Renée | 40.97 | 23.34 | 15.85 | 14.76 | 26.45 | 32.19 | 95.01 | **93.99** | **92.24** | 19.69 | 27.36 | 33.20 |
| Renée + LEVER | **41.70** | **24.76** | **17.07** | **20.38** | **37.07** | **45.13** | **95.19** | 93.91 | 92.07 | **24.76** | **32.63** | **38.15** |

Table 2: Comparison of LEVER with other tail specific XC approaches. LEVER outperforms regularization and augmentation-based methods by an average of $4\%$ in coverage and $3\%$ in PSP.

| | LF-AmazonTitles-131K | | | | | | LF-AOL-270K | | | | | |
|---|---|---|---|---|---|---|---|---|---|---|---|---|
| | C@1 | C@3 | C@5 | PSP@1 | PSP@3 | PSP@5 | C@1 | C@3 | C@5 | PSP@1 | PSP@3 | PSP@5 |
| Renée | 31.31 | 53.50 | 61.03 | 38.47 | 44.87 | 50.33 | 12.40 | 29.77 | 36.53 | 14.76 | 26.45 | 32.19 |
| Renée +TAUG | 29.47 | 51.52 | 58.68 | 36.49 | 42.83 | 47.85 | 12.46 | 29.26 | 35.88 | 15.72 | 26.74 | 32.35 |
| Renée + BoW | 30.03 | 51.78 | 59.17 | 36.96 | 42.86 | 48.09 | 12.67 | 34.32 | 43.45 | 15.58 | 30.28 | 37.90 |
| Renée + L2Reg | 31.66 | 53.65 | 60.80 | 38.74 | 44.53 | 49.49 | 8.67 | 21.07 | 26.27 | 12.21 | 20.09 | 24.36 |
| Renée + GLaS | 31.90 | 54.02 | 61.15 | 38.74 | 44.53 | 49.49 | 12.36 | 29.41 | 36.06 | 14.67 | 26.11 | 36.75 |
| Renée + Gandalf | **33.17** | **55.36** | **62.22** | **40.49** | **45.83** | **50.96** | 12.63 | 29.82 | 36.31 | 15.10 | 26.64 | 32.17 |
| Renée + LEVER | 32.50 | 54.59 | 61.42 | 39.70 | 45.44 | 50.31 | **17.43** | **42.54** | **52.01** | **20.38** | **37.07** | **45.14** |

| | LF-Wikipedia-500K | | | | | | LF-WikiHierarchy-1M | | | | | |
|---|---|---|---|---|---|---|---|---|---|---|---|---|
| Renée | 22.90 | 50.08 | 61.59 | 37.10 | 50.27 | 55.68 | 6.72 | 11.49 | 14.65 | 19.69 | 27.36 | 33.20 |
| Renée + TAUG | 19.88 | 44.74 | 56.13 | 33.76 | 46.54 | 52.16 | 3.59 | 7.19 | 9.94 | 16.95 | 24.06 | 29.69 |
| Renée + BoW | 22.92 | 49.64 | 61.40 | 36.66 | 49.79 | 55.55 | 7.84 | 14.77 | 18.39 | 24.25 | 31.10 | 36.30 |
| Renée + L2Reg | 26.52 | 53.95 | 65.14 | 39.55 | 52.42 | 57.43 | 5.61 | 9.96 | 12.98 | 18.56 | 25.90 | 31.49 |
| Renée + GLaS | 23.43 | 52.02 | 63.90 | 37.27 | 51.54 | 57.15 | 6.89 | 11.82 | 15.08 | 20.07 | 27.82 | 33.70 |
| Renée + Gandalf | 23.09 | 49.87 | 61.24 | 37.05 | 49.94 | 55.31 | 6.92 | 13.17 | 17.52 | 21.84 | 30.05 | 36.09 |
| Renée + LEVER | **29.46** | **58.53** | **70.29** | **42.93** | **55.00** | **60.29** | **9.32** | **16.41** | **20.29** | **24.76** | **32.63** | **38.15** |

the NGAME encoder for initialization, shows comparatively modest gains with LEVER. Moreover, Table 4 in the appendix illustrates the performance of LEVER when combined with XReg (Prabhu et al., 2020), an extension of Parabel, showcasing that LEVER can effectively combine with non-DNN-based methods too.

**Comparison with Tail Extreme Classification Methods**: In Table 2, we present a comparative analysis of Renée + LEVER against leading tail label-specialized methods. Note that these approaches can be easily integrated with OvA classifiers without any architectural modifications.

These methods can be broadly categorized into two classes: (1) regularization-based, such as GLaS and $L2$-regularization. GLaS promotes the proximity of classifiers for labels with similar ground truths, while $L2$-regularization introduces an additional $L2$ loss between tail expert label embeddings and label classifiers. (2) Augmentation-based, such as TAUG and Gandalf, which introduce additional training data for labels. Detailed comparisons with other prominent Extreme Classification methods, including XR-Transformer (Zhang et al., 2021), ELIAS, CascadeXML, NGAME, and ECLARE (Mittal et al., 2021b), are provided in Table 7 within the appendix. Our primary focus here is on tail label performance, hence we report PSP and coverage metrics.

LEVER consistently outperforms the second-best method by an average margin of 4% in coverage and 3% in PSP. Notably, on datasets characterized by significant skew and *multi-intent* scenarios, LEVER exhibits substantial gains in comparison to approaches like GLaS and Gandalf, which rely on ground truth data to model label correlations. For example, in the query completion task on the AOL dataset, the label *"who wrote To Kill a Mockingbird"* co-occurs with labels like *"wholesale t-shirts"* or *"who am I"* as they share the prefix *"who"*. Training classifiers with such diverse targets can lead to associations between dissimilar labels, hampering classifier training. Using Bag of Words (BoW) features from label text to model label connections alleviates the multi-intent and skew issue to some extent, as observed when Renée+ BoW performs better than Renée+ GLaS/Gandalf in LF-AOL-270K and LF-WikiHierarchy-1M. However, LEVER goes further by learning semantic associations between labels and documents through a tail-expert Siamese network, surpassing raw text-based methods.

**Comparison with Siamese Teacher**: Table 15 in the appendix compares LEVER against its corresponding Siamese encoder-based teacher. LEVER utilizes the teacher to improve OvA performance on the tail without degrading the classifier performance on the head labels. As a result, the student model in LEVER can surpass its own teacher in overall performance since it outperforms the Siamese teacher on the head labels while more-or-less equalizing on the tail.

**Comparison with an ensemble of OvA classifier and tail-expert**: To combine the strengths of OvA classifiers and encoder, another option might be to consider an ensemble model that uses predictions from the OvA model for head labels and the encoder predictions for the tail labels. Table 8 in the appendix compares LEVER with an ensemble of OvA (Renée) and Siamese Encoder. LEVER outperforms the ensemble on both precision and tail metrics. A more detailed discussion of this is provided in Sec. C.3 of the appendix.

**Choice of expert encoder**: LEVER utilizes a 6-layer DistilBert as an expert encoder. In Table 11 in the appendix we show results for two other light-weight encoders: a 3-layer MiniLM (Wang et al., 2020) and Astec Encoder (Dahiya et al., 2021b). We observe that a superior expert encoder leads to improved performance in both P and PSP.

**Effect of varying $\tau$**: Table 21 and Figure 7 in appendix shows effect of varying $\tau$ on LEVER's performance. Increasing $\tau$ improves performance on tail, while it hurts head and torso labels.

**LEVER Computational Cost**: Since LEVER is a training time-only modification, it leaves the inference costs unchanged while increasing the training time on average by 3.1x. Table 23 in the appendix shows the training time for different models and datasets when combined with LEVER. Note that in ELIAS and CascadeXML, where the train times increase by a greater margin, the gains provided by LEVER are also higher (avg. +6.1% increase in PSP and +2% increase in P). Tables 24, 25 and 26, in the appendix show the break down of the train times for Renée, ELIAS, and CascadeXML respectively.

## 6 CONCLUSIONS

This paper presented a novel approach to address the challenges of tail performance in Extreme Classification (XC) by focusing on label variance, a previously unexplored factor. It proposed LEVER framework for leveraging a tail-robust teacher model to systematically reduce label variance, thereby enhancing the performance of one-vs-all classifiers. It further developed an effective instantiation of this framework using a specialized Siamese teacher model. Experimental results on various XC datasets demonstrated significant improvements in tail performance metrics when LEVER was integrated with leading extreme classifiers, and advanced the state-of-the-art in XC. Finally, this paper also released two new and multi-intent datasets for robust benchmarking in XC.

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

CONTENTS

## A  THEORETICAL PROOFS

**Theorem 1.** Let $\mathbf{R}, \hat{\mathbf{R}}$ be the population risk and empirical risk for a binary classification loss $\mathcal{L}$. Let $\mathcal{M}_N$ be the uniform covering number (Menon et al., 2021a) corresponding to $\mathcal{L}$. Then, given the definitions established earlier, for any $\delta \in (0, 1)$, the following inequality holds with probability at least $1 - \delta$ over sampling the data points $\{\mathbf{x}\}_{i=1}^N$:

$$\mathbf{R} \leq \hat{\mathbf{R}} + \mathcal{O}\left(\sqrt{\mathbb{V}_{\mathbf{x}}[\mathcal{L}(p_x, \mathbf{w}^\top \mathbf{x})] + \mathbb{E}_{\mathbf{x}}[\mathbb{V}_y[y|\mathbf{x}]](CLWB)^2}\sqrt{\frac{\log(\mathcal{M}_N/\delta)}{N}} + \frac{\log(\mathcal{M}_N/\delta)}{N}\right) \tag{10}$$

where, $\mathbb{V}_{\mathbf{x}}\mathcal{L}(p_x, \mathbf{w}^\top \mathbf{x})$ and $\mathbb{V}_y[y|\mathbf{x}]$ are the variances in the loss function contributed by sampling of data points $\{\mathbf{x}\}_{i=1}^N$, and conditional sampling of labels $\{y\}_{i=1}^N$, respectively.

*Proof.* Applying Proposition 2. from (Menon et al., 2021a) to our setting gives the following initial result:

$$\mathbf{R} \leq \hat{\mathbf{R}} + \mathcal{O}\left(\sqrt{\mathbb{V}_{\mathbf{x},y}\mathcal{L}(y, \mathbf{w}^\top \mathbf{x})}\sqrt{\log(\mathcal{M}_N/\delta)/N} + \log(\mathcal{M}_N/\delta)/N\right) \tag{11}$$

The following simplifications can be made by leveraging basic probabilistic calculus:

$$
\begin{aligned}
\mathbb{V}_{\mathbf{x},y}[\mathcal{L}(y, \mathbf{w}^\top \mathbf{x})] &= \mathbb{V}_{\mathbf{x}}[\mathbb{E}_y[\mathcal{L}(y, \mathbf{w}^\top \mathbf{x})|\mathbf{x}]] + \mathbb{E}_{\mathbf{x}}[\mathbb{V}_y[\mathcal{L}(y, \mathbf{w}^\top \mathbf{x})|\mathbf{x}]] \ \text{(by law of total variance)} \\
&= \mathbb{V}_{\mathbf{x}}[\mathbb{E}_y[\mathcal{L}(y, \mathbf{w}^\top \mathbf{x})|\mathbf{x}]] + \mathbb{E}_{\mathbf{x}}[\mathbb{V}_y[Cyf(1, \mathbf{w}^\top \mathbf{x}) + (1-y)f(0, \mathbf{w}^\top \mathbf{x})|\mathbf{x}]] \\
&= \mathbb{V}_{\mathbf{x}}[\mathcal{L}(p_x, \mathbf{w}^\top \mathbf{x})] + \mathbb{E}_{\mathbf{x}}[\mathbb{V}_y[y|\mathbf{x}](Cf(1, \mathbf{w}^\top \mathbf{x}) - f(0, \mathbf{w}^\top \mathbf{x}))^2] \\
&= \mathbb{V}_{\mathbf{x}}[\mathcal{L}(p_x, \mathbf{w}^\top \mathbf{x})] + \mathbb{E}_{\mathbf{x}}[\mathbb{V}_y[y|\mathbf{x}]d_x^2] \\
&\text{where, } d_x = (Cf(1, \mathbf{w}^\top \mathbf{x}) - f(0, \mathbf{w}^\top \mathbf{x})) \\
d_x^2 &= C^2 f^2(1, \mathbf{w}^\top \mathbf{x}) + f^2(0, \mathbf{w}^\top \mathbf{x}) - 2C.f(1, \mathbf{w}^\top \mathbf{x}).f(0, \mathbf{w}^\top \mathbf{x}) \\
&\text{Since } f \geq 0 \text{ we get,} \\
d_x^2 &\leq C^2 f^2(1, \mathbf{w}^\top \mathbf{x}) + f^2(0, \mathbf{w}^\top \mathbf{x})
\end{aligned}
$$

(12) appears aligned with the fourth line, (13) with the last line.

Using 13 in 12 gives,

$$\mathbb{V}_{\mathbf{x},y}[\mathcal{L}(y, \mathbf{w}^\top \mathbf{x})] \leq \mathbb{V}_{\mathbf{x}}[\mathcal{L}(p_x, \mathbf{w}^\top \mathbf{x})] + \mathbb{E}_{\mathbf{x}}\mathbb{V}_y[y|\mathbf{x}](C^2 f^2(1, \mathbf{w}^\top \mathbf{x}) + f^2(0, \mathbf{w}^\top \mathbf{x}))] \tag{14}$$

Assuming $f(0,0) = f(1,0) = f_0$ (a small constant), and applying the Lipschitz continuity of $f$ gives,

$$
\begin{aligned}
|f(y, \mathbf{w}^\top \mathbf{x}) - f(y, 0)| &\leq L|\mathbf{w}^\top \mathbf{x}| \leq LWB \quad \forall y \in \{0, 1\} \\
f(y, \mathbf{w}^\top \mathbf{x}) &\leq LWB + f_0
\end{aligned} \tag{15}
$$

Using 15 in 14 gives,

$$
\begin{aligned}
\mathbb{V}_{\mathbf{x},y}[\mathcal{L}(y, \mathbf{w}^\top \mathbf{x})] &\leq \mathbb{V}_{\mathbf{x}}[\mathcal{L}(p_x, \mathbf{w}^\top \mathbf{x})] + \mathbb{E}_{\mathbf{x}}\mathbb{V}_y[y|\mathbf{x}](C^2 + 1)(LWB + f_0)^2 \\
&\approx \mathcal{O}\left(\mathbb{V}_{\mathbf{x}}[\mathcal{L}(p_x, \mathbf{w}^\top \mathbf{x})] + \mathbb{E}_{\mathbf{x}}\mathbb{V}_y[y|\mathbf{x}](CLWB)^2\right)
\end{aligned} \tag{16}
$$

Using 16 in 11 completes the proof,

$$\mathbf{R} \leq \hat{\mathbf{R}} + \mathcal{O}\left(\sqrt{\mathbb{V}_{\mathbf{x}}[\mathcal{L}(p_x, \mathbf{w}^\top \mathbf{x})] + \mathbb{E}_{\mathbf{x}}\mathbb{V}_y[y|\mathbf{x}](CLWB)^2}\sqrt{\frac{\log(\mathcal{M}_N/\delta)}{N}} + \frac{\log(\mathcal{M}_N/\delta)}{N}\right) \tag{17}$$

$\square$

**Lemma 1.** Assuming the loss weighting factor $C$ is defined in Equation 2 as $C = \frac{N}{S}$, where $S$ is the threshold defined in Equation 3 and $N$ is the number of training points, the variance term $\mathcal{V} = \mathbb{E}_{\mathbf{x}}[\mathbb{V}_y[y|\mathbf{x}]](CLWB)^2$ in Theorem 1 is bounded by $\frac{N(LWB)^2}{S}$.

*Proof.* Recall from Section 3.1 that $p_x = \mathbb{P}(y = 1|\mathbf{x})$ and $\mathbb{E}_{\mathbf{x}}[p_x] \leq \frac{S}{N}$. Using $\mathbb{V}_y[y|\mathbf{x}] = p_x(1-p_x)$ gives,

$$\mathcal{V} = \mathbb{E}_{\mathbf{x}}[p_x(1 - p_x)](N/S)^2(LWB)^2 \leq \mathbb{E}_{\mathbf{x}}[p_x]\frac{(NLWB)^2}{S^2} \leq \frac{N(LWB)^2}{S} \tag{18}$$

$\square$

**Theorem 2.** Let $\mathbf{R}, \hat{\mathbf{R}}$ be the population risk and empirical risk for a binary classification loss $\mathcal{L}$. Let $\mathcal{M}_N$ be the uniform covering number (Menon et al., 2021a) corresponding to $\mathcal{L}$. Also, let the teacher be imperfect with maximum possible error in relevance estimates bounded by $E = \|p_{\mathbf{x}} - \hat{p}_{\mathbf{x}}\|_\infty$. Then, solving the following regularized optimization problem:

$$\hat{\mathbf{R}}_s = \min_{\mathbf{w}} \frac{\lambda}{N} \sum_{i=1}^{N} \mathcal{L}(y_i, \mathbf{w}^\top \mathbf{x}_i) + \frac{1 - \lambda}{N} \sum_{i=1}^{N} \mathcal{L}(\hat{p}_i, \mathbf{w}^\top \mathbf{x}_i) \tag{19}$$

and setting $\lambda$ to minimize population risk will given the following bound for any $\delta \in (0, 1)$, the following inequality holds with probability at least $1 - \delta$ over sampling the data points $\{\mathbf{x}\}_{i=1}^{N}$:

$$\lambda = \frac{c}{b}\sqrt{\frac{a}{b^2 - c^2}} \quad ; \quad \mathbf{R} \leq \hat{\mathbf{R}}_s + \sqrt{a - a\frac{c^2}{b^2}} + c \tag{20}$$

$$\text{where,} \quad a = V_x \frac{\log(\mathcal{M}_N/\delta)}{N} \quad ; \quad b = \sqrt{S}CLWB\sqrt{\frac{\log(\mathcal{M}_N/\delta)}{N}} \quad ; \quad c = ECLWB \tag{21}$$

*Proof.* Teacher tends to be imperfect with relevance estimates $\hat{p}_x$. In this case, let us train the classifier using targets $s_x = \lambda y_x + (1 - \lambda)\hat{p}_x$. Let the corresponding population and empirical risks when trained on $s_x$ be $\mathbf{R}_s, \hat{\mathbf{R}}_s$ respectively. Then, the following holds:

$$\mathbf{R} - \hat{\mathbf{R}}_s = \mathbf{R} - \mathbf{R}_s + \mathbf{R}_s - \hat{\mathbf{R}}_s$$
$$\leq \|\mathbf{R} - \mathbf{R}_s\| + \|\mathbf{R}_s - \hat{\mathbf{R}}_s\|$$

The first term can be bounded as follows:

$$\mathbf{R} - \mathbf{R}_s = \mathbb{E}_{\mathbf{x}}\mathbb{E}_{y|\mathbf{x}} \ \mathcal{L}(y_x, \mathbf{w}^\top\mathbf{x}) - \mathcal{L}(\lambda y_x + (1 - \lambda)\hat{p}_x, \mathbf{w}^\top\mathbf{x})$$
$$= \mathbb{E}_{\mathbf{x}} \ \mathcal{L}(p_x, \mathbf{w}^\top\mathbf{x}) - \mathcal{L}(\lambda p_x + (1 - \lambda)\hat{p}_x, \mathbf{w}^\top\mathbf{x})$$
$$\text{Assuming, } \mathcal{L}(y, \mathbf{w}^\top\mathbf{x}) = Cyf(1, \mathbf{w}^\top\mathbf{x}) + (1 - y)f(0, \mathbf{w}^\top\mathbf{x})$$
$$\mathbf{R} - \mathbf{R}_s = \mathbb{E}_{\mathbf{x}}[(1 - \lambda)(p_x - \hat{p}_x)(Cf(1, \mathbf{w}^\top\mathbf{x}) - f(0, \mathbf{w}^\top\mathbf{x}))]$$
$$\leq (1 - \lambda)\|p_x - \hat{p}_x\|_\infty \max_{\mathbf{x}} (Cf(1, \mathbf{w}^\top\mathbf{x}) - f(0, \mathbf{w}^\top\mathbf{x}))$$
$$\leq (1 - \lambda)E(C + 1)(LWB + f_0)$$
$$\approx \mathcal{O}((1 - \lambda)ECLWB) \tag{22}$$

where $E$ is the upper bound over the error in the teacher's relevance estimates.

The second term $\|\mathbf{R}_s - \hat{\mathbf{R}}_s\|$ can be bounded by applying (A):

$$\mathbf{R}_s \leq \hat{\mathbf{R}}_s + \mathcal{O}\Big(\sqrt{V_x + \mathbb{E}_{\mathbf{x}}[\mathbb{V}_y[\lambda y + (1-\lambda)\hat{p}_x|\mathbf{x}]](CLWB)^2]}\sqrt{\frac{\log(\mathcal{M}_N/\delta)}{N}} + \frac{\log(\mathcal{M}_N/\delta)}{N}\Big)$$

Now, $\mathbb{V}_y[\lambda y + (1-\lambda)\hat{p}_x|\mathbf{x}] = \mathbb{V}_y[\lambda y|\mathbf{x}]$

$= \lambda^2 \mathbb{V}_y[y|\mathbf{x}]$

$$\mathbf{R}_s \leq \hat{\mathbf{R}}_s + \mathcal{O}\Big(\sqrt{V_x + \lambda^2 \frac{S(CLWB)^2}{N}}]\sqrt{\frac{\log(\mathcal{M}_N/\delta)}{N}} + \frac{\log(\mathcal{M}_N/\delta)}{N}\Big)$$

where, $\dfrac{S}{N} \approx \mathbb{E}_x p_x \geq \mathbb{E}_x \mathbb{V}_y[y|\mathbf{x}]$ (23)

As a result:

$$\mathbf{R} \leq \hat{\mathbf{R}}_s + \mathcal{O}\Big(\sqrt{V_x + \lambda^2 \frac{S(CLWB)^2}{N}}]\sqrt{\frac{\log(\mathcal{M}_N/\delta)}{N}} + \frac{\log(\mathcal{M}_N/\delta)}{N}\Big) + \mathcal{O}((1-\lambda)ECLWB) \tag{24}$$

As $\lambda$ is a regularization hyper-parameter, it value needs to be set so as to minimize the generalization error. Theoretically, this can be achieved by solving:

$$\min_{\lambda} \sqrt{a + \lambda^2 b^2} + (1-\lambda)c$$

$$\text{where,} \quad a = V_x \frac{\log(\mathcal{M}_N/\delta)}{N}$$

$$b = \sqrt{S \log(\mathcal{M}_N/\delta)} \frac{CLWB}{N}$$

$$c = ECLWB \tag{25}$$

Let's assume a reasonably small bias in teacher estimate. Specifically, let $c < b$ which means that the error due to teacher bias is relatively smaller than the error due to label variance.

Now, taking the derivative w.r.t $\lambda$ and setting it to 0, we get:

$$\lambda = \frac{c}{b}\sqrt{\frac{a}{b^2 - c^2}} \tag{26}$$

$$\sqrt{a + \lambda^2 b^2} + (1-\lambda)c = \sqrt{a - a\frac{c^2}{b^2}} + c \tag{27}$$

□

**Theorem 3.** Given a label $\mathbf{z}$, and a pair of data points $\mathbf{x}_a, \mathbf{x}_b$. Let $p_a, p_b$ be the probabilities that the label is relevant to points $a, b$ respectively. Then, assuming that (9) is fully minimized, the expected loss in (9) is minimized for $p_a = 1/(1 + e^{-(\mathbf{z}^\top \mathbf{x}_a + c)}), p_b = 1/(1 + e^{-(\mathbf{z}^\top \mathbf{x}_b + c)})$

*Proof.* The expected loss between the triplet is given by:

$$p_a(1 - p_b)\log(1 + e^{\mathbf{z}^\top \mathbf{x}_b - \mathbf{z}^\top \mathbf{x}_a}) + p_b(1 - p_a)\log(1 + e^{\mathbf{z}^\top \mathbf{x}_a - \mathbf{z}^\top \mathbf{x}_b})$$

Assuming $\Delta = \mathbf{z}^\top \mathbf{x}_b - \mathbf{z}^\top \mathbf{x}_a$ and taking the gradient w.r.t $\mathbf{z}$ gives,

$$= p_a(1 - p_b)\frac{e^{\Delta}(\mathbf{x}_b - \mathbf{x}_a)}{1 + e^{\Delta}} + p_b(1 - p_a)\frac{e^{-\Delta}(\mathbf{x}_a - \mathbf{x}_b)}{1 + e^{-\Delta}}$$

$$= \frac{\mathbf{x}_b - \mathbf{x}_a}{1 + e^{\Delta}}\Big(e^{\Delta}p_a(1 - p_b) - p_b(1 - p_a)\Big) \tag{28}$$

Setting $p_a = 1/(1 + e^{-(\mathbf{z}^\top \mathbf{x}_a + c)}), p_b = 1/(1 + e^{-(\mathbf{z}^\top \mathbf{x}_b + c)})$ in 28 gives,

$$= \frac{\mathbf{x}_b - \mathbf{x}_a}{1 + e^\Delta} \left( \frac{(e^{\mathbf{z}^\top \mathbf{x}_b - \mathbf{z}^\top \mathbf{x}_a})(e^{-(\mathbf{z}^\top \mathbf{x}_b + c)})}{(1 + e^{-(\mathbf{z}^\top \mathbf{x}_a + c)})(1 + e^{-(\mathbf{z}^\top \mathbf{x}_b + c)})} - \frac{(e^{-(\mathbf{z}^\top \mathbf{x}_a + c)})}{(1 + e^{-(\mathbf{z}^\top \mathbf{x}_a + c)})(1 + e^{-(\mathbf{z}^\top \mathbf{x}_b + c)})} \right)$$
$$= 0 \tag{29}$$

From 29 we see that the derivative of the loss is 0 when $p_a = 1/(1 + e^{-(\mathbf{z}^\top \mathbf{x}_a + c)}), p_b = 1/(1 + e^{-(\mathbf{z}^\top \mathbf{x}_b + c)})$ thus minimizing the expected loss. Note that this calibration strategy is in line with posthoc calibration strategies discussed in Platt (2000), where a model is learned, and then a parametrized sigmoid function is fit to learn the relevance probabilities. □

## B DATASET DETAILS

### B.1 DATASET STATISTICS

Table 3 shows the statistics of benchmark datasets including the newly contributed *multi-intent* datasets.

Table 3: Dataset Statistics. Pos-80% is an imbalance metric (Schultheis et al., 2022) defined as minimum fraction of class labels that retain 80% of all positive labels in the dataset. Lower value corresponds to higher skew.

| | Dataset | Train Docs | Test Docs | Labels | Avg. Labels/Doc | Avg. Docs/Label | Pos-80% |
|---|---|---|---|---|---|---|---|
| Existing | LF-AmazonTitles-131K | 294,805 | 134,835 | 131,073 | 2.29 | 5.15 | 47.5 |
| | LF-Amazon-131K | 294,805 | 134,835 | 131,073 | 2.29 | 5.15 | 47.5 |
| | LF-WikiSeeAlso-320K | 693,082 | 177,515 | 312,330 | 2.11 | 4.68 | 37.4 |
| | LF-Wikipedia-500K | 1,813,391 | 783,743 | 501,070 | 4.77 | 24.75 | 25.1 |
| | LF-AmazonTitles-1.3M | 2,248,619 | 970,237 | 1,305,265 | 22.20 | 38.24 | 28.9 |
| New | LF-AOL-270K | 3,922,479 | 519,352 | 272,825 | 2.01 | 28.83 | 11.6 |
| | LF-WikiHierarchy-1M | 1,589,378 | 397,952 | 976,214 | 25.98 | 42.31 | 7.3 |

### B.2 QK-20M DATASET

Query Keyword (QK) matching is an essential element in applications such as sponsored search. In these applications, users express their intent by querying a search engine, while advertisers bid on relevant phrases from the same domain, referred to as keywords. The retrieval (or matching system) is responsible for matching user queries to relevant advertisements. To train a query-keyword matching system, we build a dataset using click logs from Bing. We begin by considering 20M popular advertiser bid phrases (or keywords), and then corresponding to each keyword we add relevant queries to the ground truth based on whether there was a user click on the keyword when the user searched for that particular query.

### B.3 MULTI-INTENT DATASET PREPARATION

#### B.3.1 LF-AOL-270K

**Task Description**: Query auto-completion involves matching a query prefix to completing suffixes, *e.g.* given a prefix, 'cheap nike s' recommending suffix completions like 'shoes', 'shirts' etc. LF-AOL-270K is curated from AOL search logs (Pass et al., 2006) for the task of query auto-completion where (prefix, suffix) pairs are modeled as (doc, label) pairs. Retrieved suffixes from this task can be combined with user prefixes to get full query completions as proposed in (Mitra & Craswell, 2015).

**Dataset generation**: The dataset generation process involved three steps (i) Pre-processing, (ii) Prefix-suffix generation, and (iii) Post-processing.

**Pre-processing**: Queries in AOL search logs were de-duplicated and non-alphanumeric characters were removed. Queries with less than three characters were filtered since auto-completion is rarely required for those. Additionally, steps prescribed in (Kim, 2019) were followed for pre-processing and train-test splits creation.

**Prefix-suffix generation**: After pre-processing, a shortlist of the top 10M popular suffixes was derived from the train split, based on their frequency in queries. These suffixes are popular n-grams (word-level) up to 100 characters appearing at the end of queries. Sampling was done to ensure that each train query has at least one suffix from 10M suffix shortlist. Ground truth suffixes were added for sampled prefixes yielding 9.3M suffixes and 5.67M distinct prefixes (training points). Using the 10M suffix shortlist, the process of sampling prefixes was repeated in the test split, resulting in 460K suffixes.

**Post-processing**: Train-test leakage was avoided by removing all prefixes in the test set that appeared in the train set. The suffix (label) set is derived from the intersection of train and test suffixes to have a fixed label set. Finally, the dataset contains 272K labels (suffixes), 3.9M training points (prefixes), and 519K test points (test prefixes).

Code to create the dataset from raw AOL search logs is available here [1] .

### B.3.2  LF-WIKIHIERARCHY-1M

**Task description**: Taxonomy completion task involves matching a category with its generalized parent categories. LF-WikiHierarchy-1M uses Wikipedia categories to build a taxonomy completion task where documents are categories and labels are its parent categories. Articles in Wikipedia are assigned categories, which serve as semantic tags. (*e.g.* 'FIFA World Cup 2022' article has a category tag of 'Football'). These categories are arranged in a taxonomy-like structure where each category is linked to zero or more parent categories. The parent of a category is its direct generalization, *e.g.* the category 'Football' has direct parent categories 'Athletic Sports', 'Team sports' and 'Ball Games'. This taxonomy-like structure is called *Wikipedia Category graph (WCG)* and has been well studied in (Zesch & Gurevych, 2007; Benaouicha et al., 2016).

**Dataset generation** The dataset generation process involved four steps (i) Raw data collection, (ii) Pre-processing, (iii) Label set generation from WCG and (iv) Post-processing.

**Raw data collection**: The WCG is created by using the English Wikimedia dump as of 03/23 [2]. The dump contains the list of all Wikipedia categories and their links.

**Pre-processing**: To create the WCG we first filter out all meta categories used for Wikipedia maintenance, *e.g*. 'Wikipedia missing topics', 'Wikipedia new articles', 'Categories for renaming' etc. The complete list of filtered meta-categories are released as part of the code. Post filtering, the resulting WCG is a directed acyclic graph with 1,993,526 categories (nodes) and 5,781,016 edges. Each edge is a document-label pair.

**Label set generation from WCG**: The WCG in its current form only contains direct parents and misses out on potentially important ground truth information. For example, the category 'Football' will not have categories like 'Sports', 'Athletic Sports', and 'Team activities' as its labels as they are not its *direct* parents. On the other hand, adding all reachable nodes as labels leads to vague document-label pairs. For example, starting from 'Football' one can reach the category 'Cosmopolitan mammals' as follows: 'Football' → 'Athletic sports' → 'Sports by type' → 'Sports' → 'Entertainment' → 'Human activities' → 'Humans' → 'Cosmopolitan mammals'.

To maximize the relevant ground truth document-label pairs while also avoiding wrong matches like *'Football'* → *'Cosmopolitan mammals'* we limit the traversal to a maximum depth of 3 which gave the optimal trade-off (i.e. maximize true positives while avoiding false positives). Thus, in the above example, only categories up to 'Sports' are added as labels. Please refer to Fig. 1 for more clarity. Subsequently, we get 1,987,330 documents and 976,214 labels with 51,643,812 edges between them. Note that both the number of documents and labels are less than the total number of categories (1,993,526). Some categories will not have a parent and therefore won't be added as a document. Similarly, categories that are not parents of any category will not be added as labels. The final dataset is created by taking an 80%-20% random train-test split.

**Post-processing**: The dataset contains categories that occur as both documents and labels. For example, the category 'Football' occurs both as a label (for 'Football clubs', 'History of Football', etc), and as a document. Since a category will never have itself as a label we filter off pairs like

---

[1]https://github.com/anirudhb11/LEVER/tree/main/datasets/AOL
[2]https://dumps.wikimedia.org/enwiki/20230301/

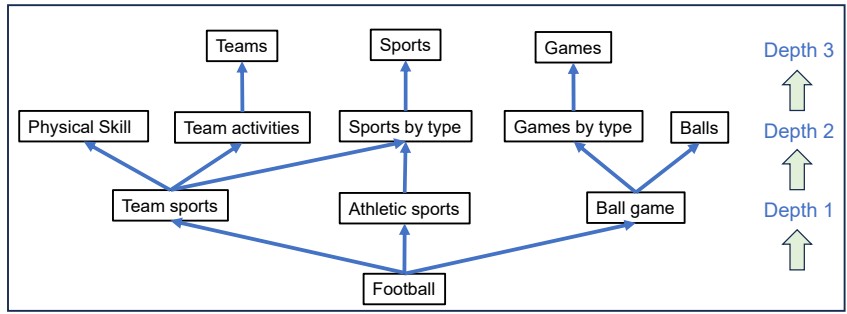

Figure 1: Snapshot of WCG graph starting from category 'Football'. All categories (nodes) reachable from 'Football' till the depth of 3 are added to its ground truth label set.

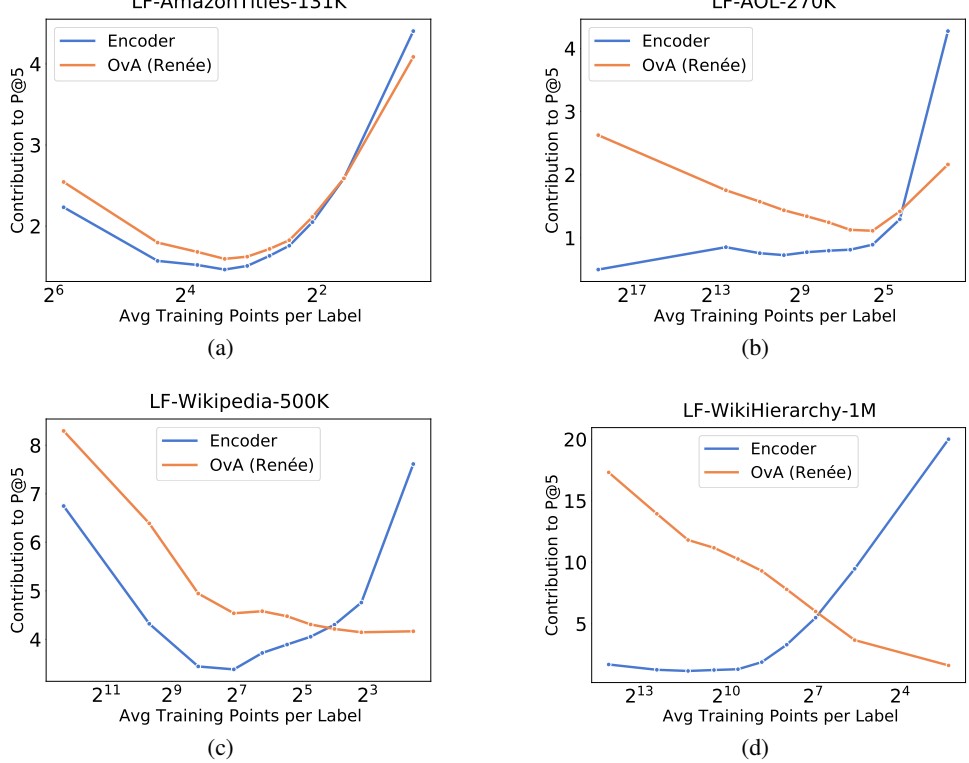

Figure 2: P@5 comparison of Siamese Encoder (blue) and OvA Classifier Renée (orange) on homogeneous (LF-AmazonTitles-131K: Fig. 2a, LF-Wikipedia-500K: Fig. 2c) and heterogeneous datasets (LF-AOL-270K: Fig. 2b, LF-WikiHierarchy-1M: Fig. 2d). Labels are partitioned into equi-volume bins based on their frequencies along the X-axis. The difference in performance (on both head and tail) is wider for heterogeneous datasets.

'Football'→'Football' during evaluation so as to not unfairly penalize Siamese-based models that rank such pairs at the top.

The LF-WikiHierarchy-1M dataset is available here [3]

---

[3]https://github.com/anirudhb11/LEVER/tree/main/datasets/WikiHierarchy

# C   ADDITIONAL RESULTS

## C.1   LEVER'S PERFORMANCE ON NON-DNN METHODS

Table 4 illustrates the performance of LEVER when combined with XReg (Prabhu et al., 2020), an extension of Parabel, showcasing that LEVER can effectively combine with non-DNN-based methods.

Table 4: Performance Comparison of XReg and XReg + LEVER on LF-AOL-270K and LF-AmazonTitles-131K

| Dataset | Model | P@1 | P@3 | P@5 | PSP@1 | PSP@3 | PSP@5 | C@1 | C@3 | C@5 |
|---|---|---|---|---|---|---|---|---|---|---|
| LF-AmazonTitles-131K | XReg | 33.1 | 22.3 | 16.0 | 24.5 | 29.4 | 33.54 | 20.22 | 36.63 | 42.84 |
| | XReg + LEVER | **38.0** | **24.7** | **17.6** | **31.4** | **35.1** | **39.1** | **25.84** | **43.47** | **49.68** |
| LF-AOL-270K | XReg | **27.0** | 14.3 | 9.9 | 7.0 | 11.0 | 14.1 | 4.18 | 10.58 | 14.44 |
| | XReg + LEVER | 26.1 | 14.3 | **10.1** | **9.2** | **17.9** | **24.0** | **6.79** | **20.59** | **28.65** |

## C.2   COMPARISON WITH SOTA AND TAIL XC METHODS

Table 5 demonstrates the enhanced performance achieved by applying LEVER to top-performing Extreme Classification (XC) methods, including ELIAS, CascadeXML, and Renée. On average PSP metrics are boosted by 5%, Coverage improves by 6.5% and Precision improves by 1.4%.

Table 7 presents a comparison of LEVER with various OvA-based methods (XR-Transformer, ELIAS, and CascadeXML) and Siamese encoder methods (NGAME and ECLARE). It's worth noting that for WikiHierarchy-1M, OvA and Siamese approaches exhibit significant trade-offs between precision and tail metrics.

## C.3   COMPARISON WITH ENSEMBLE BETWEEN TAIL EXPERT AND OvA CLASSIFIER

In the ensemble model, for each data point, the encoder and OvA model provide a shortlist of top-$k$ labels along with their prediction scores. These two shortlists (containing a total of up to $2k$ labels) need to be combined into a single shortlist of $k$ labels by tie-breaking as elaborated below. First, the labels with a frequency more than the cut-off are considered from the OvA's shortlist. Similarly, the labels with a frequency less than the cut-off are considered from the encoder's shortlist. Cut-offs are derived on the basis of the cross-over points between Encoder and Renée in the decile wise plots shown in Fig. 2. Then, the two resulting shortlists are combined by considering the assigned label scores from both models and retaining only the $k$ overall highest-scoring labels. Table 8 shows that LEVER clearly outperforms the ensemble model in 3 out of 4 datasets across all metrics. In the case of LF-WikiHierarchy-1M, the ensemble model shows gains in coverage metrics ($\sim$4-5%), this comes at the expense of a significant loss in Precision ($\sim$30%). Figure 3 compares the performance of LEVER with the ensemble model and here we see a clear dip in the torso deciles. To better understand why the ensemble curve doesn't exactly mimic the OvA curve before the cutoff and encoder curve after the cutoff, consider the following toy example:

Assume a dataset $D$ with 8 labels which are partitioned into 3 deciles (head, torso, and tail deciles). Out of 8 labels, 3 belong to the head decile ($H_1, H_2, H_3$), 2 belong to the torso decile ($O_1, O_2$) and the remaining 3 belong to the tail decile ($T_1, T_2, T_3$). The cut-off threshold partitions the label set into 2 sets: (i) labels with frequency greater than cut-off: ($H_1, H_2, H_3, O_2$) and labels with frequency less than cut-off: ($O_1, T_1, T_2, T_3$). Assume a data point $d$ has ground truth labels: ($H_1, H_2, O_1, O_2, T_1$).

Below we list the predictions of different models in the format of "label ID:model score"

Top-5 encoder predictions ($T_1 : 0.8, T_2 : 0.6, T_3 : 0.4, O_1 : 0.2, O_2 : 0.1$).

Top-5 OvA predictions ($H_1 : 0.7, H_2 : 0.5, H_3 : 0.3, O_2 : 0.2, O_1 : 0.1$)

To compute the ensemble model predictions, we first restrict the predictions of the individual models based on the cutoff frequency, i.e Encoder's predictions are restricted to ($O_1, T_1, T_2, T_3$) and OvA predictions are restricted to ($H_1, H_2, H_3, O_2$). This gives the following filtered shortlists:

Table 5: Using LEVER with leading OvA approaches improves their tail label performance consistently across benchmarks, with an average gain of 5% in PSP and 6.5% in coverage (C), while maintaining comparable precision (P) with an average gain of 1.4%.

| Model | LF-AmazonTitles-131K | | | | | | | | |
|---|---|---|---|---|---|---|---|---|---|
| | P@1 | P@3 | P@5 | PSP@1 | PSP@3 | PSP@5 | C@1 | C@3 | C@5 |
| ELIAS | 37.28 | 25.18 | 18.14 | 28.95 | 34.45 | 39.08 | 23.73 | 42.36 | 49.06 |
| ELIAS + LEVER | **42.86** | **28.37** | **20.16** | **36.30** | **41.05** | **45.43** | **29.81** | **49.88** | **56.25** |
| CascadeXML | 36.28 | 24.88 | 18.18 | 26.50 | 33.21 | 38.81 | 21.38 | 40.58 | 48.46 |
| CascadeXML + LEVER | **43.58** | **28.79** | **20.63** | **36.24** | **41.83** | **46.95** | **29.43** | **50.61** | **57.90** |
| Renée | 46.05 | 30.81 | **22.04** | 38.47 | 44.87 | **50.33** | 31.31 | 53.50 | 61.03 |
| Renée + LEVER | **46.44** | **30.83** | 21.92 | **39.70** | **45.44** | 50.31 | **32.50** | **54.59** | **61.42** |
| | LF-Amazon-131K | | | | | | | | |
| ELIAS | 43.03 | 29.27 | 21.20 | 33.49 | 40.80 | 46.76 | 27.04 | 49.10 | 57.34 |
| ELIAS + LEVER | **47.38** | **32.24** | **23.22** | **38.97** | **46.74** | **52.79** | **31.47** | **55.27** | **63.40** |
| CascadeXML | 43.76 | 29.75 | 21.58 | 34.05 | 41.69 | 47.96 | 27.30 | 50.18 | 58.81 |
| CascadeXML + LEVER | **48.24** | **32.82** | **23.73** | **39.09** | **47.55** | **54.18** | **31.26** | **55.97** | **64.81** |
| Renée | 48.05 | 32.33 | 23.26 | 39.32 | 47.10 | 53.51 | 31.49 | 55.81 | 64.61 |
| Renée + LEVER | **49.19** | **33.30** | **24.04** | **40.64** | **48.48** | **54.87** | **32.39** | **56.81** | **65.20** |
| | LF-WikiSeeAlso-320K | | | | | | | | |
| ELIAS | 41.40 | 27.36 | 20.66 | 23.83 | 28.38 | 31.90 | 13.30 | 27.72 | 35.50 |
| ELIAS + LEVER | **45.99** | **30.28** | **22.78** | **30.00** | **34.16** | **37.52** | **16.56** | **33.06** | **41.34** |
| CascadeXML | 30.21 | 18.72 | 14.05 | 12.46 | 14.15 | 16.25 | 6.70 | 13.38 | 17.72 |
| CascadeXML + LEVER | **38.84** | **25.43** | **19.36** | **21.62** | **25.85** | **29.45** | **12.01** | **25.12** | **32.59** |
| Renée | 47.79 | **31.73** | **23.82** | 31.13 | 36.49 | 40.37 | 17.02 | 35.32 | 44.56 |
| Renée + LEVER | **47.89** | 31.52 | 23.53 | **32.44** | **37.45** | **40.99** | **17.78** | **36.33** | **45.31** |
| | LF-Wikipedia-500K | | | | | | | | |
| ELIAS | 81.94 | 62.71 | 48.75 | 33.58 | 43.92 | 48.67 | 19.62 | 41.30 | 51.36 |
| ELIAS + LEVER | **82.44** | **63.88** | **50.03** | **36.94** | **49.28** | **55.03** | **23.55** | **50.81** | **63.04** |
| CascadeXML | 77.00 | 58.30 | 45.10 | 31.25 | 39.35 | 43.29 | 15.78 | 33.07 | 41.46 |
| CascadeXML + LEVER | **80.10** | **60.41** | **46.44** | **36.79** | **46.65** | **50.99** | **23.99** | **49.16** | **60.13** |
| Renée | 84.95 | 66.25 | 51.68 | 37.10 | 50.27 | 55.68 | 22.90 | 50.08 | 61.59 |
| Renée + LEVER | **85.02** | **66.37** | **51.98** | **42.93** | **55.00** | **60.29** | **29.46** | **58.53** | **70.29** |
| | LF-AOL-270K | | | | | | | | |
| ELIAS | 40.83 | 22.33 | 14.91 | 13.29 | 21.46 | 25.22 | 10.46 | 22.85 | 27.06 |
| ELIAS + LEVER | **40.85** | **22.83** | **15.57** | **13.68** | **24.30** | **30.43** | **10.52** | **26.33** | **33.56** |
| CascadeXML | **41.20** | **22.12** | 14.82 | **12.58** | 19.53 | 23.19 | **8.73** | 19.47 | 23.47 |
| CascadeXML + LEVER | 39.41 | 21.78 | **14.99** | 11.96 | **21.30** | **27.59** | 7.86 | **22.11** | **29.89** |
| Renée | 40.97 | 23.34 | 15.85 | 14.76 | 26.45 | 32.19 | 12.40 | 29.77 | 36.53 |
| Renée + LEVER | **41.70** | **24.76** | **17.07** | **20.38** | **37.07** | **45.13** | **17.43** | **42.54** | **52.01** |
| | LF-WikiHierarchy-1M | | | | | | | | |
| ELIAS | **95.27** | **94.25** | **92.45** | 17.15 | 24.41 | 30.01 | 4.00 | 7.78 | 10.49 |
| ELIAS + LEVER | 94.02 | 91.97 | 89.50 | **28.27** | **36.80** | **42.13** | **10.78** | **18.88** | **23.03** |
| CascadeXML | **94.88** | **93.69** | **91.79** | 16.03 | 22.87 | 28.17 | 3.12 | 6.17 | 8.52 |
| CascadeXML + LEVER | 94.77 | 93.54 | 91.56 | **20.14** | **27.49** | **33.01** | **6.68** | **11.22** | **14.13** |
| Renée | 95.01 | **93.99** | **92.24** | 19.69 | 27.36 | 33.20 | 6.72 | 11.49 | 14.65 |
| Renée + LEVER | **95.19** | 93.90 | 92.07 | **24.76** | **32.63** | **38.15** | **9.32** | **16.14** | **20.29** |
| | LF-AmazonTitles-1.3M | | | | | | | | |
| ELIAS | 47.48 | 42.21 | 38.60 | 18.79 | 23.20 | 26.06 | 11.53 | 21.45 | 27.33 |
| ELIAS + LEVER | **48.91** | **43.17** | **39.28** | **23.68** | **27.43** | **29.72** | **15.10** | **26.65** | **32.84** |
| CascadeXML | 47.14 | 41.43 | 37.73 | 15.92 | 20.23 | 23.16 | 8.65 | 16.75 | 21.95 |
| CascadeXML + LEVER | **47.98** | **42.02** | **38.12** | **20.06** | **24.51** | **27.28** | **12.36** | **22.57** | **28.52** |
| Renée | **56.10** | **49.91** | **45.32** | 28.56 | 33.38 | 36.14 | 17.31 | 30.60 | 37.59 |
| Renée + LEVER | 56.01 | 49.43 | 44.85 | **33.55** | **36.82** | **38.81** | **21.03** | **35.70** | **42.78** |

Encoder: $(T_1 : 0.8, T_2 : 0.6, T_3 : 0.4, O_1 : 0.2)$

OvA: $(H_1 : 0.7, H_2 : 0.5, H_3 : 0.3, O_2 : 0.2)$

Next, we combine and sort the labels based on the scores from both the encoder and OvA as follows:

$(T_1 : 0.8, H_1 : 0.7, T_2 : 0.6, H_2 : 0.5, T_3 : 0.4, H_3 : 0.3, O_2 : 0.2, O_1 : 0.2)$

Finally, we retain only the top-5 highest scoring labels as our final ensemble predictions:

Table 6: Using LEVER with leading OvA approaches improves their tail performance consistently across benchmarks in Macro-F1 ($+4.3\%$ on avg.), Macro-precision ($+4.1\%$ on avg.), and Macro-Recall ($+5.49\%$ on avg.). Following Zhang et al. (2023), $k$ values for datasets were chosen based on average labels per point for the dataset. We use $k$=3 for LF-AmazonTitles-131K, LF-Amazon-131K, LF-WikiSeeAlso-320K, LF-AOL-270K, $k$=5 for LF-Wikipedia-500K, $k$=25 for LF-AmazonTitles-1.3M and LF-WikiHierarchy-1M.

| Model | LF-AmazonTitles-131K | | | LF-Amazon-131K | | | LF-Wikipedia-500K | | |
|---|---|---|---|---|---|---|---|---|---|
| | F1@k | P@k | R@k | F1@k | P@k | R@k | F1@k | P@k | R@k |
| ELIAS | 24.23 | 22.82 | 31.70 | 28.48 | 26.46 | 37.93 | 27.98 | 27.97 | 35.17 |
| ELIAS + LEVER | **29.53** | **27.70** | **38.21** | **33.36** | **31.16** | **43.35** | **34.12** | **33.88** | **44.68** |
| CascadeXML | 22.23 | 20.80 | 30.36 | 28.89 | 26.74 | 38.83 | 20.04 | 20.13 | 26.75 |
| CascadeXML + LEVER | **29.12** | **26.91** | **39.04** | **33.29** | **30.76** | **44.13** | **31.40** | **31.61** | **41.70** |
| Renée | 32.19 | 30.36 | 41.55 | 32.55 | 29.95 | 43.88 | 35.94 | 36.78 | 43.76 |
| Renée + LEVER | **33.35** | **31.78** | **42.45** | **35.69** | **34.05** | **45.04** | **40.54** | **40.71** | **51.36** |

| Model | LF-AmazonTitles-1.3M | | | LF-AOL-270K | | | LF-WikiHierarchy-1M | | |
|---|---|---|---|---|---|---|---|---|---|
| | F1@k | P@k | R@k | F1@k | P@k | R@k | F1@k | P@k | R@k |
| ELIAS | 16.43 | 15.73 | 24.68 | 10.69 | 9.45 | 16.18 | 20.71 | 23.26 | 22.43 |
| ELIAS + LEVER | **18.85** | **17.92** | **28.58** | **13.97** | **12.87** | **19.41** | **26.99** | **28.75** | **30.53** |
| CascadeXML | 13.58 | 13.23 | 22.06 | 9.00 | 8.25 | 13.46 | 17.73 | 20.91 | 19.22 |
| CascadeXML + LEVER | **16.02** | **15.07** | **27.02** | **11.59** | **11.03** | **15.74** | **21.17** | **24.36** | **22.85** |
| Renée | 24.79 | 24.16 | 35.42 | 17.70 | 16.88 | 22.53 | 24.48 | 27.59 | 26.04 |
| Renée + LEVER | **26.72** | **25.73** | **37.04** | **22.38** | **20.40** | **30.18** | **27.89** | **30.67** | **30.23** |

| Model | LF-WikiSeeAlso-320K | | |
|---|---|---|---|
| | F1@k | P@k | R@k |
| ELIAS | 17.67 | 16.83 | 22.46 |
| ELIAS + LEVER | **22.06** | **21.21** | **27.26** |
| CascadeXML | 5.86 | 5.20 | 9.78 |
| CascadeXML + LEVER | **14.81** | **13.90** | **20.00** |
| Renée | 23.28 | 22.17 | 29.30 |
| Renée + LEVER | **23.79** | **22.70** | **30.12** |

Ensemble predictions: $(T_1 : 0.8, H_1 : 0.7, T_2 : 0.6, H_2 : 0.5, T_3 : 0.4)$ Table 9 shows the contribution to P@5 for different models across the three deciles. Note that the example is in line with our observations in Fig. 3 where (i) Encoder performs better on tail deciles (blue curve), (ii) OvA models perform better on head deciles (orange), (iii) Ensemble (green) between Encoder and OvA models perform comparably to Encoders on tail deciles and OvA based models on head deciles but incurs significant losses in torso deciles, (iv) Performance of the ensemble model can be worse than the individual models (e.g. ensemble P@5 < OvA P@5 in toy example). If the Ensemble model were to dominate both Encoder and OvA models it should have achieved decile-wise contributions of $(2/5, 2/5, 1/5)$ which is not the case. On average, the torso labels are ranked relatively lower by both models since neither model specializes in them. Further, when combined using the proposed ensemble these labels get more aggressively down-voted.

## C.4 Effect of Re-ranking on LEVER and other Tail XC approaches

Table 10 illustrates the impact of post-hoc reranking using inverse propensity scores, on LEVER and other Tail XC approaches. The application of reranking shows varying degrees of trade-offs between precision and tail metrics across different models. Notably, while LEVER attains superior performance in tail metrics for three out of four datasets, there is a trade-off in precision compared to other methods in the LF-WikiHierarchy-1M dataset.

## C.5 Ablations

**Effect of Teacher Model**: Table 11 demonstrates the impact of employing various encoders as a tail expert. We conduct a comparison with two alternative encoders: (i) MiniLM, a 3-layer transformer

Table 7: Comparison between LEVER and leading OvA-based methods such as XR-Transformer, ELIAS, and CascadeXML, as well as Siamese encoder-based methods like NGAME and ECLARE. Note that for LF-WikiHierarchy-1M, OvA and Siamese-based methods display significant trade-offs between precision and tail metrics. Siamese-based methods score much higher in PSP numbers (+16 on average) but lag behind in precision (-14 on average) when compared to OvA-based methods.

| | | P@1 | P@3 | P@5 | PSP@1 | PSP@3 | PSP@5 | C@1 | C@3 | C@5 |
|---|---|---|---|---|---|---|---|---|---|---|
| | | LF-AmazonTitles-131K | | | | | | | | |
| SOTA XC Methods | XR-Transformer | 38.10 | 25.57 | 18.32 | 28.86 | 34.85 | 39.59 | 20.24 | 40.70 | 48.87 |
| | ELIAS | 37.28 | 25.18 | 18.14 | 28.95 | 34.45 | 39.08 | 23.73 | 42.36 | 49.06 |
| | CascadeXML | 36.28 | 24.88 | 18.18 | 26.50 | 33.21 | 38.81 | 21.38 | 40.58 | 48.46 |
| | ECLARE | 41.40 | 27.58 | 19.82 | 34.22 | 39.69 | 44.63 | 27.91 | 48.38 | 55.48 |
| | NGAME | **46.58** | 30.41 | 21.49 | 39.54 | 44.77 | 49.59 | 32.35 | 53.78 | 60.73 |
| | Renée | 46.05 | 30.81 | **22.04** | 38.47 | 44.87 | 50.33 | 31.31 | 53.50 | 61.03 |
| Tail XC Methods | Renée +TAUG | 44.34 | 29.73 | 21.15 | 36.49 | 42.83 | 47.85 | 29.47 | 51.52 | 58.68 |
| | Renée + BoW | 42.95 | 29.18 | 21.03 | 36.96 | 42.86 | 48.09 | 30.03 | 51.78 | 59.17 |
| | Renée + L2Reg | 45.19 | 29.92 | 21.29 | 38.47 | 44.23 | 49.24 | 31.66 | 53.65 | 60.80 |
| | Renée + GLaS | 45.35 | 30.03 | 21.33 | 38.74 | 44.53 | 49.49 | 31.90 | 54.02 | 61.15 |
| | Renée + Gandalf | 45.86 | 30.53 | 21.79 | **40.49** | **45.83** | **50.96** | **33.17** | **55.36** | **62.22** |
| | Renée + LEVER | 46.44 | **30.83** | 21.92 | 39.70 | 45.44 | 50.31 | 32.50 | 54.59 | 61.42 |
| | | LF-AOL-270K | | | | | | | | |
| SOTA XC Methods | XR-Transformer | 37.56 | 20.44 | 13.94 | 11.76 | 21.10 | 26.31 | 8.83 | 23.07 | 29.44 |
| | ELIAS | 40.83 | 22.33 | 14.91 | 13.29 | 21.46 | 25.22 | 10.46 | 22.85 | 27.06 |
| | CascadeXML | 41.20 | 22.12 | 14.82 | 12.58 | 19.53 | 23.19 | 8.73 | 19.47 | 23.74 |
| | ECLARE | 28.53 | 16.18 | 11.55 | 10.11 | 18.69 | 24.58 | 7.41 | 20.59 | 28.11 |
| | NGAME | 39.44 | 22.29 | 15.30 | 16.33 | 29.63 | 37.06 | 14.28 | 34.70 | 43.84 |
| | Renée | 40.97 | 23.34 | 15.85 | 14.76 | 26.45 | 32.19 | 12.40 | 29.77 | 36.53 |
| Tail XC Methods | Renée +TAUG | 40.40 | 22.80 | 15.56 | 15.72 | 26.74 | 32.35 | 12.46 | 29.26 | 35.88 |
| | Renée + BoW | 41.11 | 23.91 | 16.41 | 15.58 | 30.28 | 37.90 | 12.67 | 34.32 | 43.45 |
| | Renée + L2Reg | 39.83 | 21.75 | 14.71 | 12.21 | 20.09 | 24.36 | 8.67 | 21.07 | 26.27 |
| | Renée + GLaS | 40.91 | 23.25 | 15.78 | 14.67 | 26.11 | 31.75 | 12.36 | 29.41 | 36.06 |
| | Renée + Gandalf | 40.63 | 23.01 | 15.58 | 15.10 | 26.64 | 32.17 | 12.63 | 29.82 | 36.31 |
| | Renée + LEVER | **41.71** | **24.77** | **17.07** | **20.38** | **37.07** | **45.14** | **17.43** | **42.54** | **52.01** |
| | | LF-Wikipedia-500K | | | | | | | | |
| SOTA XC Methods | XR-Transformer | 81.62 | 61.38 | 47.85 | 33.58 | 42.97 | 47.81 | 19.05 | 40.05 | 50.66 |
| | ELIAS | 81.94 | 62.71 | 48.75 | 33.58 | 43.92 | 48.67 | 19.62 | 41.30 | 51.36 |
| | CascadeXML | 77.00 | 58.3 | 45.10 | 31.25 | 39.35 | 43.29 | 15.78 | 33.07 | 41.46 |
| | NGAME | 84.32 | 65.59 | 51.41 | 39.88 | 50.74 | 57.09 | 26.22 | 51.42 | 64.79 |
| | Renée | 84.95 | 66.25 | 51.68 | 37.10 | 50.27 | 55.68 | 22.90 | 50.08 | 61.59 |
| Tail XC Methods | Renée +TAUG | 83.07 | 64.46 | 50.32 | 33.76 | 46.54 | 52.16 | 19.88 | 44.74 | 56.13 |
| | Renée + BoW | 84.43 | 66.09 | 51.74 | 36.66 | 49.79 | 55.55 | 22.92 | 49.64 | 61.40 |
| | Renée + L2Reg | 84.57 | 66.05 | 51.50 | 39.55 | 52.42 | 57.43 | 26.52 | 53.95 | 65.14 |
| | Renée + GLaS | 84.85 | **66.63** | **52.09** | 37.27 | 51.54 | 57.15 | 23.43 | 52.02 | 63.90 |
| | Renée + Gandalf | 84.59 | 66.07 | 51.63 | 37.05 | 49.94 | 55.31 | 23.09 | 49.87 | 61.24 |
| | Renée + LEVER | **85.02** | 66.37 | 51.98 | **42.93** | **55.00** | **60.29** | **29.46** | **58.53** | **70.29** |
| | | LF-WikiHierarchy-1M | | | | | | | | |
| SOTA XC Methods | XR-Transformer | 95.33 | **94.26** | **92.39** | 15.96 | 23.04 | 28.62 | 2.98 | 6.23 | 8.86 |
| | ELIAS | 95.27 | 94.25 | 92.45 | 17.15 | 24.41 | 30.01 | 4.00 | 7.78 | 10.49 |
| | CascadeXML | 94.88 | 93.69 | 91.79 | 16.03 | 22.87 | 28.17 | 3.12 | 6.17 | 8.52 |
| | ECLARE | 90.95 | 89.14 | 86.90 | 15.70 | 22.41 | 27.65 | 2.57 | 5.94 | 9.30 |
| | NGAME | 83.16 | 78.24 | 73.90 | **38.43** | **44.22** | **47.93** | 7.83 | **22.59** | **29.25** |
| | Renée | 95.01 | 93.99 | 92.24 | 19.69 | 27.36 | 33.20 | 6.72 | 11.49 | 14.65 |
| Tail XC Methods | Renée +TAUG | **95.34** | 94.45 | 92.27 | 16.95 | 24.06 | 29.69 | 3.59 | 7.19 | 9.94 |
| | Renée + BoW | 93.92 | 92.04 | 90.27 | 24.25 | 31.10 | 36.30 | 7.84 | 14.77 | 18.39 |
| | Renée + L2Reg | 94.68 | 93.45 | 91.56 | 18.56 | 25.90 | 31.49 | 5.61 | 9.96 | 12.98 |
| | Renée + GLaS | 95.01 | 93.98 | 92.26 | 20.07 | 27.82 | 33.70 | 6.89 | 11.82 | 15.08 |
| | Renée + Gandalf | 93.01 | 90.85 | 88.16 | 21.84 | 30.05 | 36.09 | 6.92 | 13.17 | 17.52 |
| | Renée + LEVER | 95.19 | 93.90 | 92.07 | 24.76 | 32.63 | 38.15 | **9.32** | 16.41 | 20.29 |

model, and (ii) Astec, which learns a projection matrix from sparse Bag of Words (BoW) features to a dense embedding space. The results highlight that the choice of a superior teacher substantially enhances the performance of LEVER.

**Effect of sampling strategy:** LEVER makes use of NGAME Module (Dahiya et al., 2023a) trained using mini-batches of labels instead of documents. The modification helps specialize the Siamese

Table 8: Comparison of LEVER with an ensemble of OvA and tail expert encoder. LEVER outperforms the ensemble consistently on all metrics for 3 out of 4 datasets . Note that for LF-WikiHierarchy-1M even though the ensemble improves coverage, the drop in precision is very large (31% on average).

|  | P@1 | P@3 | P@5 | PSP@1 | PSP@3 | PSP@5 | C@1 | C@3 | C@5 |
|---|---|---|---|---|---|---|---|---|---|
| **LF-AmazonTitles-131K** | | | | | | | | | |
| Ensemble | 42.98 | 26.84 | 18.23 | 37.24 | 41.58 | 44.69 | 30.38 | 51.57 | 57.54 |
| Renée + LEVER | **46.44** | **30.83** | **21.92** | **39.70** | **45.44** | **50.31** | **32.82** | **55.11** | **61.94** |
| **LF-AOL-270K** | | | | | | | | | |
| Ensemble | 35.20 | 20.33 | 13.69 | 19.43 | 36.98 | 44.74 | 17.19 | 44.80 | 54.87 |
| Renée + LEVER | **41.71** | **24.77** | **17.07** | **20.38** | **37.07** | **45.14** | **17.43** | 42.54 | 52.01 |
| **LF-Wikipedia-500K** | | | | | | | | | |
| Ensemble | 82.55 | 61.96 | 46.65 | 39.82 | 51.29 | 55.76 | 25.72 | 58.46 | **72.17** |
| Renée + LEVER | **85.02** | **66.42** | **52.05** | **42.50** | **54.86** | **60.20** | **29.46** | **58.53** | 70.29 |
| **LF-WikiHierarchy-1M** | | | | | | | | | |
| Ensemble | 67.48 | 62.65 | 58.39 | **28.08** | 31.75 | 34.01 | **10.86** | **20.82** | **25.89** |
| Renée + LEVER | **95.19** | **93.90** | **92.07** | 24.79 | **32.74** | **38.29** | 9.08 | 16.12 | 20.02 |

Table 9: P@5 performance for different models across deciles.

| Model | Head Decile P@5 | Torso Decile P@5 | Tail Decile P@5 | Overall P@5 |
|---|---|---|---|---|
| Encoder | 0/5 | 2/5 | 1/5 | 3/5 |
| OvA | 2/5 | 2/5 | 0/5 | 4/5 |
| Ensemble | 2/5 | 0/5 | 1/5 | 3/5 |

Table 10: Performance comparison of LEVER with other tail XC approaches LEVER outperforms other tail XC methods in tail metrics on 3 out of 4 datasets. while LEVER attains superior performance in tail metrics for three out of four datasets, there is a trade-off in precision compared to other methods in the LF-WikiHierarchy-1M dataset.

| Dataset | Model | P@1 | P@3 | P@5 | Ps@1 | Ps@3 | Ps@5 | C@1 | C@3 | C@5 |
|---|---|---|---|---|---|---|---|---|---|---|
| LF-AmazonTitles-131K | Renée | 46.05 | 30.81 | 22.04 | 38.47 | 44.87 | 50.33 | 31.31 | 53.50 | 61.03 |
|  | + Rerank | **46.16** | **30.80** | **22.02** | 39.99 | 45.53 | 50.78 | 32.90 | 54.65 | 61.84 |
|  | + L2Reg + ReRank | 44.89 | 29.71 | 21.14 | 39.99 | 44.58 | 49.29 | 33.18 | 54.48 | 61.19 |
|  | + GLaS + ReRank | 45.06 | 29.82 | 21.17 | 40.18 | 44.83 | 49.49 | 33.36 | 54.78 | 61.48 |
|  | + Gandalf + ReRank | 44.17 | 30.29 | 21.90 | 40.98 | **46.09** | **51.19** | 33.61 | **56.27** | **62.97** |
|  | + LEVER + ReRank | 45.36 | 30.67 | 21.95 | **41.13** | 46.00 | 50.85 | **33.91** | 55.79 | 62.31 |
| LF-AOL-270K | Renée | 40.97 | 23.34 | 15.85 | 15.06 | 26.36 | 31.97 | 12.40 | 29.77 | 36.53 |
|  | + Rerank | **41.53** | 24.11 | 16.44 | 20.21 | 31.11 | 37.24 | 20.27 | 36.13 | 43.01 |
|  | + L2Reg + ReRank | 40.25 | 22.43 | 15.25 | 15.16 | 23.59 | 28.81 | 13.69 | 26.38 | 32.56 |
|  | + GLaS + ReRank | 41.41 | 24.01 | 16.37 | 19.93 | 30.71 | 36.82 | 20.05 | 35.71 | 42.56 |
|  | + Gandalf + ReRank | 40.87 | 23.49 | 15.94 | 20.70 | 30.68 | 36.22 | 20.61 | 35.47 | 41.65 |
|  | + LEVER + ReRank | 39.60 | **24.23** | **16.92** | **28.20** | **41.58** | **49.33** | **27.40** | **48.95** | **57.62** |
| LF-Wikipedia-500K | Renée | 84.95 | 66.25 | 51.68 | 37.10 | 50.27 | 55.68 | 22.90 | 50.08 | 61.59 |
|  | + Rerank | 79.28 | 63.56 | 50.80 | 53.44 | 56.16 | 59.06 | 41.58 | 59.52 | 67.17 |
|  | + L2Reg + ReRank | 79.30 | 63.64 | 50.69 | 57.67 | 58.06 | 60.32 | 45.03 | 62.90 | 70.14 |
|  | + GLaS + ReRank | 80.20 | **64.74** | **51.50** | 53.22 | 56.85 | 60.07 | 41.49 | 60.17 | 68.55 |
|  | + Gandalf + ReRank | **80.58** | 64.55 | 51.29 | 51.03 | 55.09 | 58.36 | 39.07 | 58.00 | 66.23 |
|  | + LEVER + ReRank | 75.34 | 62.07 | 50.26 | **59.15** | **60.29** | **62.95** | **47.24** | **68.40** | **75.94** |
| LF-WikiHierarchy-1M | Renée | 95.01 | 93.99 | 92.24 | 19.69 | 27.36 | 33.20 | 6.62 | 11.39 | 14.56 |
|  | + Rerank | 89.95 | 89.94 | 88.86 | 44.15 | 52.89 | 58.47 | 18.15 | 30.53 | 35.16 |
|  | + L2Reg + ReRank | 91.18 | 90.92 | 89.51 | 41.22 | 49.41 | 55.00 | 16.34 | 27.78 | 32.73 |
|  | + GLaS + ReRank | **93.09** | **92.42** | **91.00** | 46.38 | 54.75 | 60.17 | 18.78 | 31.41 | 36.07 |
|  | + Gandalf + ReRank | 86.03 | 83.04 | 80.88 | 51.77 | 57.79 | 61.37 | **20.21** | 34.82 | 39.73 |
|  | + LEVER + ReRank | 86.27 | 84.51 | 83.69 | **52.17** | **58.92** | **63.59** | 19.60 | **34.84** | **40.58** |

encoder towards tail labels. Renée + LEVER$_{doc}$ denotes the model that uses NGAME encoder with mini-batches of documents to augment the training data. Table 12 shows the effect of sampling strategy by comparing Renée + LEVER$_{doc}$ and Renée + LEVER. Renée + LEVER outperforms Renée + LEVER$_{doc}$ by upto 2% in PSP while being comparable in precision.

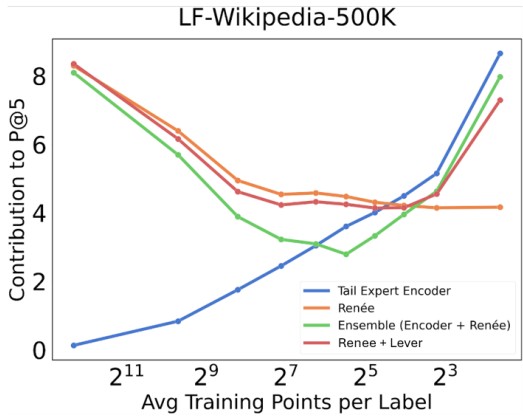

Figure 3: Performance comparison of a Tail Expert Encoder (blue), an OvA Classifier (orange), an Ensemble of Expert Encoder and OvA Classifier (Green) and LEVER-based OvA Classifier (red) in the presence of label skew. Labels are partitioned into equi-volume bins based on their frequencies along the X-axis. OvA overfits to tail labels with few training points. Encoder leverages label metadata to improve on tail but underfits to head. Ensemble mode suffers on torso labels. LEVER combines the strengths of both OvA and Encoder to perform well on all labels. The macro prefix has been omitted for the sake of brevity.

Table 11: Comparison of different encoders: a 3-layer MiniLM and Astec, and their effects on LEVER performance. The Astec encoder learns a projection matrix that maps sparse Bag-of-Words features to a dense embedding space. A superior teacher leads to improved performance in both Precision and tail metrics, namely PSP and coverage. Note that for all we add the same number of neighbours for each label across all teachers.

| | P@1 | P@3 | P@5 | PSP@1 | PSP@3 | PSP@5 | C@1 | C@3 | C@5 |
|---|---|---|---|---|---|---|---|---|---|
| | | | | **LF-AmazonTitles-131K** | | | | | |
| Astec Encoder | 19.78 | 18.28 | 14.39 | 16.96 | 27.38 | 33.61 | 14.34 | 35.77 | 44.37 |
| MinLM-L3 Encoder | 23.86 | 21.65 | 16.82 | 20.22 | 32.44 | 39.26 | 17.20 | 41.80 | 50.82 |
| DistilBERT-L6 Encoder | 41.33 | 28.71 | 20.77 | 39.24 | 44.62 | 49.52 | 32.83 | 55.11 | 61.95 |
| Renée | 46.05 | 30.81 | 22.04 | 38.47 | 44.87 | 50.33 | 31.31 | 53.50 | 61.03 |
| Renée + LEVER (Astec) | 42.76 | 28.97 | 20.97 | 36.09 | 42.25 | 47.82 | 29.54 | 51.29 | 59.01 |
| Renée + LEVER (MiniLM-L3) | 45.26 | 30.40 | 21.82 | 38.28 | 44.67 | 50.09 | 31.22 | 53.73 | 61.11 |
| Renée + LEVER (DistilBERT-L6) | 46.44 | 30.83 | 21.92 | 39.70 | 45.44 | 50.31 | 32.82 | 55.11 | 61.94 |
| | | | | **LF-AmazonTitles-1.3M** | | | | | |
| Astec Encoder | 36.14 | 30.25 | 26.32 | 28.12 | 29.00 | 29.29 | 18.38 | 31.56 | 37.83 |
| MiniLM-L3 Encoder | 32.10 | 26.86 | 23.43 | 25.48 | 26.20 | 26.48 | 16.81 | 29.32 | 35.46 |
| DistilBERT-L6 Encoder | 42.27 | 36.16 | 31.63 | 35.62 | 38.11 | 38.87 | 22.37 | 38.93 | 46.98 |
| Renée | 56.10 | 49.91 | 45.32 | 28.56 | 33.38 | 36.14 | 17.61 | 30.60 | 37.59 |
| Renée + LEVER (Astec) | 49.30 | 43.12 | 39.26 | 30.46 | 33.83 | 35.86 | 18.39 | 33.10 | 40.71 |
| Renée + LEVER (MiniLM-L3) | 50.24 | 44.01 | 40.08 | 32.73 | 35.90 | 37.73 | 20.09 | 35.55 | 43.23 |
| Renée + LEVER (DistilBERT-L6) | 56.01 | 49.43 | 44.85 | 33.55 | 36.82 | 38.81 | 21.03 | 35.70 | 42.78 |

**Effect of varying $\tau$:** The hyperparameter $\tau$ is tuned using a validation set that contains 5% of the training data. The best value of $\tau$ obtained is then used to train LEVER on complete training data. Fig. 7 shows the effect of varying $\tau$ on LEVER's performance. It can be seen that increasing $\tau$ leads to better performance on tail labels, while it hurts the head or torso labels.

**Effect of varying $\lambda$:** The hyperparameter $\lambda$ controls the importance between the two loss terms. Table 14 shows the effect of varying $\lambda$, and Figures 4, 5 and 6 show their corresponding decile-wise plots.

# D    MODEL DETAILS AND HYPERPARAMETERS

## D.1    TAIL EXPERT SIAMESE ENCODER

NGAME's (Dahiya et al., 2023a) hyperparameters include:

Table 12: Siamese teacher models, when trained with a document-wise sampling strategy (Siamese Encoder$_{doc}$) perform better in P@1 by 3% on average but are inferior in PSP@1 by 10% compared to label-wise trained teachers (Siamese Encoder$_{lbl}$). However, OvA classifiers, when distilled from Siamese Encoder$_{lbl}$ (Renée + LEVER$_{lbl}$), are comparable in Precision while more accurate in PSP@1 by 1.7% compared to OvA classifiers distilled from Siamese Encoder$_{doc}$ (Renée + LEVER$_{doc}$), thus highlighting the importance of a tail-specialized Siamese teacher model.

| Dataset | Model | P@1 | P@3 | P@5 | PSP@1 | PSP@3 | PSP@5 |
|---|---|---|---|---|---|---|---|
| LF-AmazonTitles-131K | Siamese Encoder$_{doc}$ | 43.13 | 28.99 | 20.73 | 38.72 | 43.93 | 48.81 |
| | Siamese Encoder$_{lbl}$ | 41.31 | 28.70 | 20.77 | 39.21 | 44.61 | 49.51 |
| | Renée + LEVER$_{doc}$ | 46.05 | 30.81 | **22.04** | 38.47 | 44.87 | **50.33** |
| | Renée + LEVER$_{lbl}$ | **46.44** | **30.83** | 21.92 | **39.70** | **45.44** | 50.31 |
| LF-Wikipedia-500K | Siamese Encoder$_{doc}$ | 81.96 | 60.72 | 46.23 | 48.76 | 55.87 | 58.25 |
| | Siamese Encoder$_{lbl}$ | 67.82 | 45.66 | 34.31 | **60.77** | **57.26** | 57.20 |
| | Renée + LEVER$_{doc}$ | **85.09** | 65.94 | 51.69 | 40.17 | 52.65 | 58.18 |
| | Renée + LEVER$_{lbl}$ | 85.02 | **66.37** | **51.98** | 42.93 | 55.00 | **60.29** |
| LF-WikiHierarchy-1M | Siamese Encoder$_{doc}$ | 59.08 | 53.96 | 49.58 | 50.82 | 50.20 | 50.16 |
| | Siamese Encoder$_{lbl}$ | 66.82 | 60.64 | 55.42 | **75.63** | **73.02** | **70.52** |
| | Renée + LEVER$_{doc}$ | 95.02 | **94.06** | **92.28** | 23.64 | 31.28 | 36.89 |
| | Renée + LEVER$_{lbl}$ | **95.19** | 93.90 | 92.07 | 24.79 | 32.74 | 38.29 |
| LF-AmazonTitles-1.3M | Siamese Encoder$_{doc}$ | 45.83 | 39.94 | 35.48 | 33.04 | 35.64 | 36.80 |
| | Siamese Encoder$_{lbl}$ | 42.27 | 36.16 | 31.63 | **35.62** | **38.11** | **38.87** |
| | Renée + LEVER$_{doc}$ | 55.02 | 48.94 | 44.82 | 31.86 | 36.42 | 38.75 |
| | Renée + LEVER$_{lbl}$ | **56.01** | **49.43** | **44.85** | 33.55 | 36.82 | 38.81 |

Table 13: P and PSP Comparison of NGAME, Renée, and Renée + LEVER on QK-20M Dataset

| | P@1 | P@3 | P@5 | PSP@1 | PSP@3 | PSP@5 |
|---|---|---|---|---|---|---|
| NGAME | 69.94 | 52.72 | 44.81 | 48.24 | 55.63 | 58.71 |
| Renée | **72.14** | **54.87** | **47.02** | 50.75 | 58.90 | 62.56 |
| Renée + LEVER | 71.70 | 54.48 | 46.56 | **54.74** | **63.36** | **67.14** |

- `cluster-sz`: Mini-batches in NGAME are created from clusters of similar documents (or labels). To build a batch of $B$ documents (or labels) we pick $B/$`cluster-sz` clusters.
- `cluster-freq`: Denotes the frequency of refreshing the clusters using updated embeddings.
- $\gamma$: Denotes the margin enforced while training with contrastive loss.
- `lr`: Learning rate for the encoder.
- `bsz`: Denotes the size of mini-batches.
- `epochs`: Denotes the number of epochs for which the NGAME module is trained.

To train the tail-expert NGAME module we closely follow the settings from (Dahiya et al., 2023a). NGAME utilizes a 6-layer DistilBERT architecture. Table 16 shows the hyperparameters used on benchmark as well as newly contributed datasets.

## D.2 ELIAS

ELIAS's (Gupta et al., 2022) hyperparameters include:

- $C$: Denotes the number of clusters in the index graph.
- $\alpha$: Multiplicative hyperparameter that controls the effective number of clusters that can get activated for a given input get activated for a given input.
- $\beta$: Multiplicative hyperparameter that controls the effective number of labels that can get assigned to a particular cluster.
- $\rho$: Controls the row-wise sparsity of the adjacency matrix.
- $\lambda_{elias}$: Controls importance of classification loss $\mathcal{L}_c$ and shortlist loss $\mathcal{L}_s$ in the final loss.
- $K$: Denotes the shortlist size, label classifiers are only evaluated on top-K shortlisted labels.
- $b$: Denotes the beam size.

Table 14: Effect of varying $\lambda$ when Renée is combined with LEVER. The equal weightage (0.5) gives the best performance. Increasing $\lambda$ weighs the hard labels more, resulting in performance that gets closer to the base classifier, i.e., PSP worsens, and Precision remains more or less unaffected. Decreasing $\lambda$ also helps only up to a certain point, i.e., $\lambda$=0.5; we believe this is because our teacher is not perfect, and we strike a balance between hard and soft labels. Figures 4, 5 and 6 show the decile-wise plots corresponding to these values. Note that for LF-AmazonTitles-131K the effect of varying $\lambda$ is minimal.

| | | | | LF-AmazonTitles-131K | | | | | |
|---|---|---|---|---|---|---|---|---|---|
| $\lambda$ | P@1 | P@3 | P@5 | PSP@1 | PSP@3 | PSP@5 | C@1 | C@3 | C@5 |
| 0.33 | 46.15 | 30.76 | 21.87 | 39.7 | 45.33 | 50.13 | 32.56 | 54.48 | 61.24 |
| 0.50 | 46.44 | 30.83 | 21.92 | 39.70 | 45.44 | 50.31 | 32.82 | 55.11 | 61.94 |
| 0.66 | 46.57 | 30.87 | 21.93 | 39.59 | 45.46 | 50.36 | 32.31 | 54.49 | 61.48 |
| 0.80 | 46.58 | 30.88 | 21.92 | 39.37 | 45.39 | 50.33 | 32.06 | 54.38 | 61.42 |

| | | | | LF-Wikipedia-500K | | | | | |
|---|---|---|---|---|---|---|---|---|---|
| $\lambda$ | P@1 | P@3 | P@5 | PSP@1 | PSP@3 | PSP@5 | C@1 | C@3 | C@5 |
| 0.33 | 84.66 | 65.94 | 51.54 | 40.51 | 53.4 | 58.75 | 26.88 | 56.01 | 67.94 |
| 0.50 | 85.02 | 66.42 | 52.05 | 42.50 | 54.86 | 60.20 | 29.46 | 58.53 | 70.29 |
| 0.66 | 84.96 | 66.51 | 52.18 | 39.34 | 53.53 | 59.46 | 25.66 | 55.78 | 68.33 |
| 0.80 | 84.84 | 66.42 | 52.14 | 38.66 | 53.09 | 59.16 | 24.96 | 55.08 | 67.79 |

| | | | | LF-AOL-270K | | | | | |
|---|---|---|---|---|---|---|---|---|---|
| $\lambda$ | P@1 | P@3 | P@5 | PSP@1 | PSP@3 | PSP@5 | C@1 | C@3 | C@5 |
| 0.33 | 41.14 | 24 | 16.5 | 17.24 | 32.31 | 40.02 | 14.14 | 36.67 | 45.84 |
| 0.50 | 41.70 | 24.78 | 17.07 | 20.38 | 37.07 | 45.13 | 17.43 | 42.54 | 52.01 |
| 0.66 | 41.44 | 24.41 | 16.76 | 16.76 | 32.69 | 40.47 | 14.16 | 37.38 | 46.54 |
| 0.80 | 41.17 | 24.04 | 16.49 | 15.59 | 30.3 | 37.64 | 13.16 | 34.62 | 43.25 |

Table 15: Renée (OvA) and Siamese trained encoder exhibit different trade-offs on in precision and tail-metrics (PSP, Coverage). LEVER improves the tail performance of Renée (+5% on average in PSP and +3% on average in coverage) while retaining comparable precision.

| | P@1 | P@3 | P@5 | PSP@1 | PSP@3 | PSP@5 | C@1 | C@3 | C@5 |
|---|---|---|---|---|---|---|---|---|---|
| | | | | LF-AmazonTitles-131K | | | | | |
| Siamese Encoder | 41.33 | 28.71 | 20.77 | 39.24 | 44.62 | 49.52 | **32.83** | **55.11** | **61.95** |
| Renée | 46.05 | 30.81 | **22.04** | 38.47 | 44.87 | **50.33** | 31.31 | 53.50 | 61.03 |
| Renée + LEVER | **46.44** | **30.83** | 21.92 | **39.70** | **45.44** | 50.31 | 32.50 | 54.59 | 61.42 |
| | | | | LF-AOL-270K | | | | | |
| Siamese Encoder | 23.24 | 15.67 | 11.68 | **25.41** | **36.24** | 43.43 | **27.49** | **47.45** | **55.58** |
| Renée | 40.97 | 23.34 | 15.85 | 14.76 | 26.45 | 32.19 | 12.40 | 29.77 | 36.53 |
| Renée + LEVER | **41.71** | **24.77** | **17.07** | 20.38 | 37.07 | **45.14** | 17.43 | 42.54 | 52.01 |
| | | | | LF-Wikipedia-500K | | | | | |
| Siamese Encoder | 67.81 | 45.65 | 34.31 | **60.76** | **57.25** | 57.20 | **48.76** | **71.41** | **78.59** |
| Renée | 84.95 | 66.25 | 51.68 | 37.10 | 50.27 | 55.68 | 22.90 | 50.08 | 61.59 |
| Renée + LEVER | **85.02** | **66.37** | **51.98** | 42.93 | 55.00 | **60.29** | 29.46 | 58.53 | 70.29 |
| | | | | LF-WikiHierarchy-1M | | | | | |
| Siamese Encoder | 66.82 | 60.64 | 55.42 | **75.63** | **73.02** | **70.54** | **21.82** | **42.93** | **50.62** |
| Renée | 95.01 | **93.99** | **92.24** | 19.69 | 27.36 | 33.20 | 6.72 | 11.49 | 14.65 |
| Renée + LEVER | **95.19** | 93.90 | 92.07 | 24.76 | 32.63 | 38.15 | 9.32 | 16.14 | 20.29 |

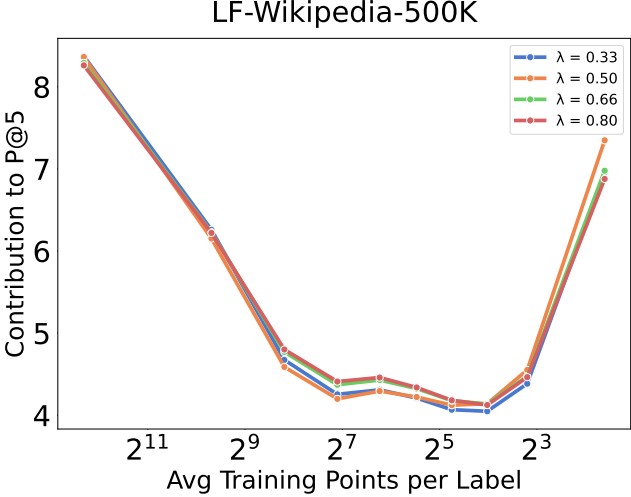

Figure 4: Effect of varying hyperparameter $\lambda$ on LEVER's head and tail performance on LF-Wikipedia-500K

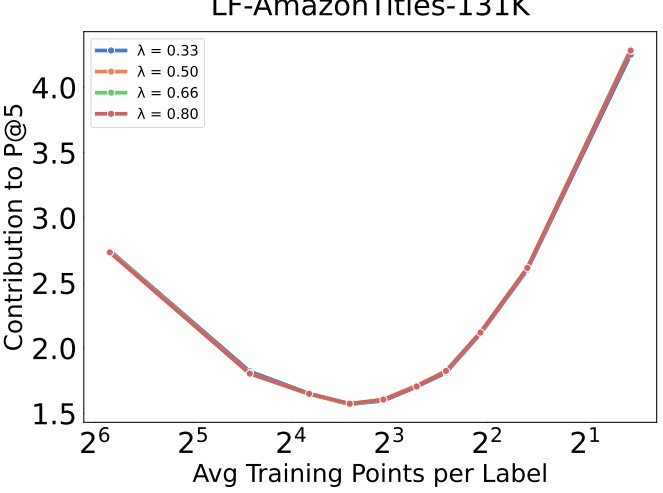

Figure 5: Effect of varying hyperparameter $\lambda$ on LEVER's head and tail performance on LF-AmazonTitles-127K. Here the choice of $\lambda$ has minimal affect on the final performance of LEVER therefore all four plots are closely superimposed.

- `epochs`: Denotes the total number of epochs (i.e. including stage 1 and stage 2 training).
- $LR_\phi, LR_W$: Denotes the learning rate used for the transformer encoder and the rest of the model.
- `bsz`: denotes the batch-size of the mini-batches used during training

We closely follow the setting used in (Gupta et al., 2022). ELIAS uses a 6-layer Distil-BERT encoder. Note that the NGAME encoder is only used to augment the ground truth with labels similar to a particular label, it is not used in any other way while training ELIAS. Table 17 shows the hyperparameters used on the benchmark as well as newly contributed datasets.

D.3 CASCADEXML

CascadeXML's (Kharbanda et al., 2022) hyperparameters include:

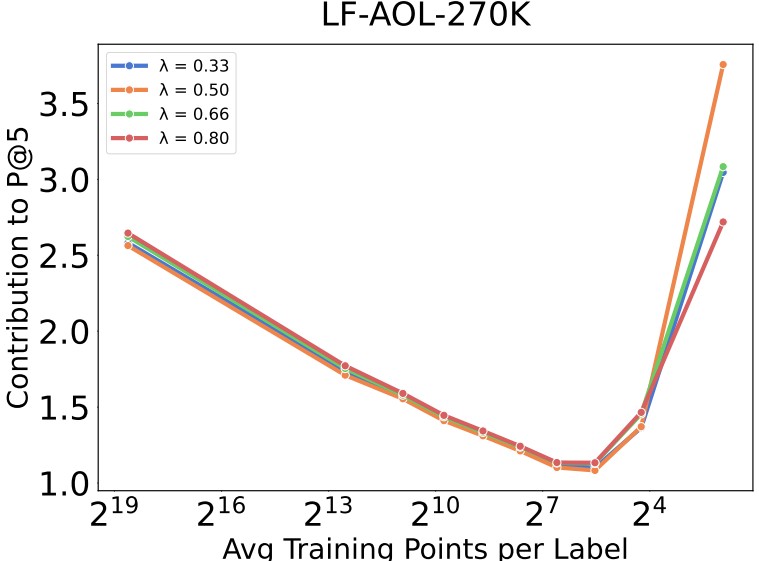

Figure 6: Effect of varying hyperparameter $\lambda$ on LEVER's head and tail performance on LF-AOL-270K

Table 16: Hyperparameters of tail-expert NGAME module. ✗ indicates use of random mini-batches.

| Dataset | cluster-sz | cluster-freq | $\gamma$ | LR | bsz | Epochs |
|---|---|---|---|---|---|---|
| LF-AmazonTitles-131K | 8 | 5 | 0.3 | $2 \times 10^{-4}$ | 1600 | 300 |
| LF-Amazon-131K | 512 | 5 | 0.3 | $2 \times 10^{-4}$ | 700 | 400 |
| LF-AOL-270K | ✗ | ✗ | 0.05 | $2 \times 10^{-4}$ | 3200 | 300 |
| LF-WikiSeeAlso-320K | 512 | 5 | 0.3 | $2 \times 10^{-4}$ | 1024 | 300 |
| LF-Wikipedia-500K | 16 | 5 | 0.3 | $2 \times 10^{-4}$ | 512 | 40 |
| LF-WikiHierarchy-1M | 1024 | 5 | 0.3 | $2 \times 10^{-4}$ | 6400 | 300 |
| LF-AmazonTitles-1.3M | 8 | 5 | 0.3 | $2 \times 10^{-4}$ | 1600 | 400 |

- `Ep`: Number of epochs CascadeXML is trained for.
- `bsz`: Denotes the batch size used for training.
- `label resolution`: Denotes the BERT layers and clustering size used at each resolution.
- `dropout`: Dropout used at each resolution.
- `shortlist size`: Cluster size used at each resolution.
- $LR_\phi, LR_W$: Denotes the learning rate used for the transformer encoder and weight vectors.

We closely follow the setting used in (Kharbanda et al., 2022). CascadeXML uses a 12-layer BERT encoder. Note that the NGAME encoder is only used to augment the ground truth with labels similar to a particular label, it is not used in any other way while training CascadeXML. Table 18 shows the hyperparameters used on benchmark as well as newly contributed datasets.

### D.4 RENÉE

Renée's (Jain et al., 2023) hyperparameters include:

- `epochs`: Denotes the total number of epochs for which Renée is trained.
- `dropout`: Denotes the probability of randomly dropping the encoder outputs in order to regularise the network.
- `warmup`: Warmup steps is the number of training iterations over which both the encoder and the classifier learning rates are linearly increased from 0 to the maximum value.

Table 17: Hyperparameters of ELIAS

| Dataset | $C$ | $\alpha$ | $\beta$ | $\rho$ | $\lambda_{elias}$ | $K$ | $b$ | Epochs | $LR_\phi$ | $LR_W$ | bsz |
|---|---|---|---|---|---|---|---|---|---|---|---|
| LF-AmazonTitles-131K | 2048 | 10 | 150 | 1000 | 0.05 | 2000 | 20 | 60 | $1 \times 10^{-4}$ | $2 \times 10^{-2}$ | 512 |
| LF-Amazon-131K | 2048 | 10 | 150 | 1000 | 0.05 | 2000 | 20 | 70 | $7 \times 10^{-5}$ | $5 \times 10^{-3}$ | 1024 |
| LF-AOL-270K | 4096 | 10 | 150 | 1000 | 0.05 | 2000 | 20 | 70 | $3 \times 10^{-5}$ | $1 \times 10^{-3}$ | 8192 |
| LF-WikiSeeAlso-320K | 4096 | 10 | 150 | 1000 | 0.05 | 2000 | 20 | 40 | $5 \times 10^{-5}$ | $5 \times 10^{-3}$ | 1024 |
| LF-Wikipedia-500K | 8192 | 10 | 150 | 1000 | 0.05 | 2000 | 20 | 40 | $5 \times 10^{-5}$ | $5 \times 10^{-3}$ | 256 |
| LF-WikiHierarchy-1M | 16384 | 10 | 150 | 1000 | 0.05 | 2000 | 20 | 30 | $5 \times 10^{-5}$ | $5 \times 10^{-3}$ | 1024 |
| LF-AmazonTitles-1.3M | 16384 | 10 | 150 | 1000 | 0.05 | 2000 | 20 | 40 | $2 \times 10^{-5}$ | $1 \times 10^{-3}$ | 1024 |

Table 18: Hyperparameters of CascadeXML

| Dataset | Ep | bsz | Label Resolution | Dropout | Shortlist-sz | $LR_\phi$ | $LR_W$ |
|---|---|---|---|---|---|---|---|
| LF-AmazonTitles-131K | 15 | 64 | $\{5,6\}:2^{10} - \{8\}:2^{13} - \{10\}:2^{16} - 12 : 131073$ | 0.2, 0.25, 0.35, 0.5 | $2^{10}, 2^{10}, 2^{10}$ | $1e^{-4}$ | $1e^{-3}$ |
| LF-Amazon-131K | 15 | 64 | $\{5,6\}:2^{9} - \{8\}:2^{12} - \{10\}:2^{15} - 12 : 131073$ | 0.2, 0.25, 0.4, 0.5 | $2^{6}, 2^{7}, 2^{8}$ | $1e^{-4}$ | $1e^{-3}$ |
| LF-AOL-270K | 12 | 96 | $\{5,6\}:2^{10} - \{8\}:2^{13} - \{10\}:2^{16} - 12 : 272825$ | 0.2, 0.25, 0.35, 0.5 | $2^{10}, 2^{10}, 2^{10}$ | $1e^{-4}$ | $1e^{-3}$ |
| LF-WikiSeeAlso-320K | 12 | 64 | $\{5,6\}:2^{10} - \{8\}:2^{13} - \{10\}:2^{16} - 12 : 312330$ | 0.2, 0.25, 0.35, 0.5 | $2^{10}, 2^{11}, 2^{12}$ | $1e^{-4}$ | $1e^{-3}$ |
| LF-Wikipedia-500K | 12 | 256 | $\{5,6\}:2^{10} - \{8\}:2^{13} - \{10\}:2^{16} - 12 : 501070$ | 0.2, 0.25, 0.35, 0.5 | $2^{10}, 2^{10}, 2^{11}$ | $1e^{-4}$ | $1e^{-3}$ |
| LF-WikiHierarchy-1M | 12 | 96 | $\{5,6\}:2^{10} - \{8\}:2^{13} - \{10\}:2^{16} - 12 : 976214$ | 0.2, 0.25, 0.35, 0.5 | $2^{10}, 2^{10}, 2^{10}$ | $1e^{-4}$ | $1e^{-3}$ |
| LF-AmazonTitles-1.3M | 10 | 48 | $\{7,8\}:2^{13} - \{10\}:2^{16} - 12 : 1305265$ | 0.2, 0.3, 0.4 | $2^{10}, 2^{11}$ | $1e^{-4}$ | $1e^{-3}$ |

- $LR_\phi, LR_W$: Denotes the learning rate used for the transformer encoder and the classifier layer.
- `bsz`: Denotes the batch size of the mini-batches used during training.
- `clf-wd`: Weight decay for fully connected layer parameters.

Table 19: Hyperparameters of Renée

| Dataset | Epochs | Dropout | Warmup | $LR_\phi$ | $LR_W$ | bsz | clf-wd |
|---|---|---|---|---|---|---|---|
| LF-AmazonTitles-131K | 100 | 0.85 | 5000 | $1 \times 10^{-5}$ | $5 \times 10^{-2}$ | 512 | $1 \times 10^{-4}$ |
| LF-Amazon-131K | 100 | 0.85 | 5000 | $1 \times 10^{-5}$ | $5 \times 10^{-2}$ | 512 | $1 \times 10^{-4}$ |
| LF-AOL-270K | 100 | 0.60 | 20000 | $1 \times 10^{-6}$ | $1 \times 10^{-3}$ | 1024 | $1 \times 10^{-4}$ |
| LF-WikiSeeAlso-320K | 100 | 0.75 | 5000 | $2 \times 10^{-4}$ | $2 \times 10^{-1}$ | 2048 | $1 \times 10^{-4}$ |
| LF-Wikipedia-500K | 100 | 0.70 | 5000 | $5 \times 10^{-5}$ | $4 \times 10^{-3}$ | 2048 | $1 \times 10^{-4}$ |
| LF-WikiHierarchy-1M | 100 | 0.70 | 20000 | $1 \times 10^{-4}$ | $2 \times 10^{-3}$ | 1024 | $1 \times 10^{-2}$ |
| LF-AmazonTitles-1.3M | 100 | 0.70 | 15000 | $1 \times 10^{-6}$ | $1 \times 10^{-2}$ | 1024 | $1 \times 10^{-4}$ |

We closely follow the setting used in (Jain et al., 2023). Renée uses a 6-layer Distil-BERT encoder. Table 19 shows the hyperparameters used on benchmark as well as newly contributed datasets.

### D.5 ReRank + TAug

ReRank + TAug (Wei et al., 2021) hyperparameters include:

- $\epsilon_{split}$: Denotes the proportion of labels that will be considered as head labels. The original dataset $D$ containing $L$ labels is split into 2 datasets $D_h$ and $D_t$. $D_h$ contains headmost $\epsilon_{split}L$ labels and their associated training points, while $D_t$ contains the remaining $L - \epsilon_{split}L$ labels along with their associated training points.
- `n-aug`: Denotes the number of additional data points that will be generated for each data point in $D_t$.
- $p_{drop}$: Denotes the probability of dropping a token from the data point.
- $p_{swap}$: Denotes the probability of swapping two randomly chosen tokens.
- `rerank-strategy`: Denotes the multiplicative factor used to re-rank scores. We use the label inverse propensity factor to perform re-ranking.

### D.6 Gandalf

Gandalf's (Kharbanda et al., 2024) hyperparameters include:

Table 20: Hyperparameters of Re-rank + TAUG

| Dataset | $\epsilon_{\text{split}}$ | n-aug | $p_{\text{drop}}$ | $p_{\text{swap}}$ |
|---|---|---|---|---|
| LF-AmazonTitles-131K | 0.90 | 8 | 0.30 | 0.30 |
| LF-AOL-270K | 0.65 | 6 | 0.20 | 0.20 |
| LF-Wikipedia-500K | 0.90 | 4 | 0.10 | 0.10 |
| LF-WikiHierarchy-1M | 0.90 | 4 | 0.20 | 0.20 |

- threshold: Denotes the threshold used to filter out labels obtained from the normalized label correlation graph during augmentation.

We closely follow the settings used in (Kharbanda et al., 2024) and use threshold of 0.1 for all datasets.

## D.7 LEVER

### D.7.1 HYPERPARAMETERS

LEVER uses the parameter $\tau$ to control the number of entities (data points or labels) that are added for each label. The hyper-parameter $c$ from Theorem 3 was set to 0, as this value worked consistently well across datasets. Table 22 displays the values of $\tau$, $\tau_d$, and $\tau_l$ for both benchmark and newly contributed datasets. We observed that setting $\tau_d = 0$ yields satisfactory results for most cases, with the exception of LF-Wikipedia-500K, for which we set $\tau_d = 12$. A value of $\lambda = 0.5$ was employed for all experiments. Table 21 illustrates the effect of varying $\tau$, and Table 14 demonstrates the effect of varying $\lambda$.

Table 21: Performance variation in P, PSP and Coverage when as $\tau$ is varied in LEVER for LF-Wikipedia-500K and LF-WikiHierarchy-1M. Figure 7 shows the decile-wise plots corresponding to these values. As more neighbours are added, the tail metrics (PSP and Coverage) improve while the Precision remains constant or slightly drops.

| | **LF-Wikipedia-500K** | | | | | | | | | |
|---|---|---|---|---|---|---|---|---|---|---|
| $\tau$ | PPL | P@1 | P@3 | P@5 | PSP@1 | PSP@3 | PSP@5 | C@1 | C@3 | C@5 |
| 1 | 18.1 | 84.96 | 66.26 | 51.68 | 37.10 | 50.27 | 55.68 | 22.90 | 50.08 | 61.59 |
| 20 | 39.5 | 85.14 | 66.90 | 52.35 | 38.39 | 52.33 | 58.04 | 24.72 | 53.45 | 65.43 |
| 45 | 64.5 | 85.02 | 66.43 | 52.05 | 42.51 | 54.86 | 60.20 | 29.08 | 58.28 | 70.09 |
| | **LF-WikiHierarchy-1M** | | | | | | | | | |
| $\tau$ | PPL | P@1 | P@3 | P@5 | PSP@1 | PSP@3 | PSP@5 | C@1 | C@3 | C@5 |
| 1 | 43.3 | 95.01 | 93.99 | 92.24 | 19.69 | 27.36 | 33.20 | 6.62 | 11.39 | 14.56 |
| 4 | 44.8 | 95.19 | 93.90 | 92.07 | 24.79 | 32.74 | 38.29 | 9.08 | 16.12 | 20.02 |
| 8 | 47.7 | 94.97 | 93.18 | 90.98 | 26.37 | 34.99 | 40.51 | 10.08 | 18.34 | 22.74 |

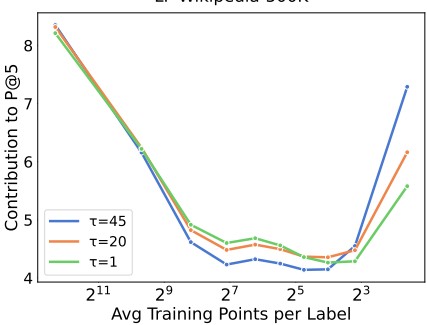
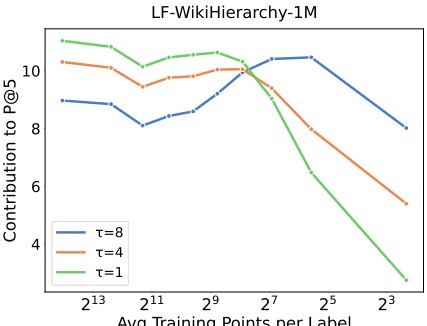

Figure 7: Effect of varying hyperparameter $\tau$ on LEVER's head and tail performance on LF-Wikipedia-500K and LF-WikiHierarchy-1M.

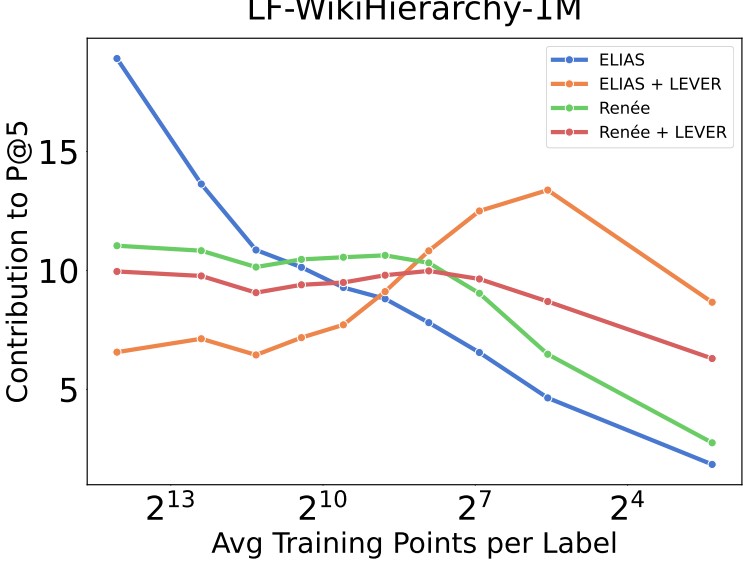

Figure 8: In ELIAS there is a bigger drop in head performance as compared to Renée when LEVER is applied. This therefore translates to a much larger increase in ELIAS+LEVER's tail performance for this dataset, note that for ELIAS the performance improvement starts from the torso labels itself.

Table 22: LEVER's Hyperparameter $\tau$ on benchmark datasets

| Dataset | $\tau$ | $\tau_l$ | $\tau_d$ |
|---|---|---|---|
| LF-AmazonTitles-131K | 15 | 15 | 0 |
| LF-Amazon-131K | 20 | 20 | 0 |
| LF-AOL-270K | 100 | 100 | 0 |
| LF-WikiSeeAlso-320K | 4 | 4 | 0 |
| LF-Wikipedia-500K | 75 | 60 | 15 |
| LF-WikiHierarchy-1M | 4 | 4 | 0 |
| LF-AmazonTitles-1.3M | 15 | 15 | 0 |

### D.7.2 TRAINING TIME

Table 23 shows the training time for different models and datasets when combined with LEVER. Note that in ELIAS and CascadeXML, where the train times increase by a greater margin, the gains provided by LEVER are also higher (avg. +6.1% increase in PSP and +2

Table 23: Training time (in hours) for different models on a single NVIDIA V100 GPU. The average training time increases by 3.1x, and in the worst case, by 8.9x. For LEVER counterparts, this includes the time for training the teacher model, generating soft labels, and using/training the OvA classifier.

| Dataset | Renée | Renée+ LEVER | ELIAS | ELIAS+ LEVER | CascadeXML | CascadeXML+ LEVER |
|---|---|---|---|---|---|---|
| LF-AmazonTitles-131K | 17.59 | 20.11 | 4.33 | 18.82 | 3.63 | 17.14 |
| LF-Amazon-131K | 42.77 | 46.51 | 19.44 | 65.62 | 4.60 | 41.31 |
| LF-AOL-270K | 136.22 | 138.20 | 60.67 | 171.20 | 42.12 | 152.00 |
| LF-WikiSeeAlso-320K | 86.42 | 95.19 | 25.33 | 111.46 | 12.40 | 88.52 |
| LF-Wikipedia-500K | 154.93 | 184.05 | 138.67 | 226.72 | 29.58 | 89.72 |
| LF-WikiHierarchy-1M | 31.44 | 40.15 | 24.00 | 61.17 | 9.85 | 35.91 |
| LF-AmazonTitles-1.3M | 154.39 | 186.45 | 40.00 | 158.23 | 70.00 | 202.93 |
| Average Time Inc. | | 1.14x | | 3.29x | | 4.86x |

Table 24, 25, 26 show the runtime break down when LEVER. Note that the training time of the Siamese Teacher is less than what is reported in (Dahiya et al., 2023a) as that includes the time taken to train both the NGAME Encoder and NGAME Classifier.

Table 24: Renée + LEVER training time (in hrs) on a single NVIDIA V100 GPU. Training LEVER involves three steps **(a)**: Time to train the Siamese teacher, **(b)**: Time to construct soft labels and **(c)**: Time to train the Renée.

| Dataset | Siamese Teacher | Soft Labels | Renée | Total |
|---|---|---|---|---|
| LF-AmazonTitles-131K | 11.82 | 0.07 | 8.22 | 20.11 |
| LF-Amazon-131K | 34.44 | 0.07 | 12.00 | 46.51 |
| LF-AOL-270K | 108.00 | 0.20 | 30.00 | 138.20 |
| LF-WikiSeeAlso-320K | 69.86 | 0.25 | 25.07 | 95.19 |
| LF-Wikipedia-500K | 50.26 | 0.45 | 133.33 | 184.05 |
| LF-WikiHierarchy-1M | 19.33 | 1.04 | 19.78 | 40.15 |
| LF-AmazonTitles-1.3M | 93.50 | 0.73 | 92.22 | 186.45 |

Table 25: ELIAS + LEVER training time (in hrs) on a single NVIDIA V100 GPU. Training LEVER involves three steps **(a)**: Time to train the Siamese teacher, **(b)**: Time to construct soft labels and **(c)**: Time to train the ELIAS.

| Dataset | Siamese Teacher | Soft Labels | ELIAS | Total |
|---|---|---|---|---|
| LF-AmazonTitles-131K | 11.82 | 0.07 | 6.93 | 18.82 |
| LF-Amazon-131K | 34.44 | 0.07 | 31.11 | 65.62 |
| LF-AOL-270K | 108.00 | 0.20 | 63.00 | 171.20 |
| LF-WikiSeeAlso-320K | 69.86 | 0.25 | 41.33 | 111.46 |
| LF-Wikipedia-500K | 50.26 | 0.45 | 176.00 | 226.72 |
| LF-WikiHierarchy-1M | 19.33 | 1.04 | 40.80 | 61.17 |
| LF-AmazonTitles-1.3M | 93.50 | 0.73 | 64.00 | 158.23 |

Table 26: CascadeXML + LEVER training time (in hrs) on a single NVIDIA V100 GPU. Training LEVER involves three steps **(a)**: Time to train the Siamese teacher, **(b)**: Time to construct soft labels and **(c)**: Time to train the CascadeXML.

| Dataset | Siamese Teacher | Soft Labels | CascadeXML | Total |
|---|---|---|---|---|
| LF-AmazonTitles-131K | 11.82 | 0.07 | 5.25 | 17.14 |
| LF-Amazon-131K | 34.44 | 0.07 | 6.80 | 41.31 |
| LF-AOL-270K | 108.00 | 0.20 | 43.80 | 152.00 |
| LF-WikiSeeAlso-320K | 69.86 | 0.25 | 18.40 | 88.52 |
| LF-Wikipedia-500K | 50.26 | 0.45 | 39.00 | 89.72 |
| LF-WikiHierarchy-1M | 19.33 | 1.04 | 15.54 | 35.91 |
| LF-AmazonTitles-1.3M | 93.50 | 0.73 | 108.70 | 202.93 |

