# OpenReview forum: "Enhancing Tail Performance in Extreme Classifiers by Label Variance Reduction"
_ICLR.cc/2024/Conference — ICLR 2024 poster_

### Official Review · Reviewer_UBHy · 2023-10-18

**Soundness:** 3 good
**Presentation:** 1 poor
**Contribution:** 2 fair
**Rating:** 5
**Confidence:** 2

**Summary:**

This paper investigates a novel factor, label variance, in the context of tail performance in extreme classifiers. To address this issue, the authors propose LEVER, a knowledge distillation framework that leverages a robust teacher model to reduce label variance. Experimental results demonstrate that LEVER significantly improves tail performance, achieving approximately 5% and 6% increases in PSP and Coverage metrics, respectively, when integrated with state-of-the-art extreme classifiers.

**Strengths:**

1. The paper give a proof of the correlation between the generalization performance of a classifier and the variance in labels.
2. Use a Siamese- style model as a teacher to help reduce the label variance effects.
3. The extensive experiment results are promising.
4. The paper contribute two new datasets to the field.

**Weaknesses:**

1. The readability of the paper is not strong, and the formatting is uncomfortable. For example, the abstract should generally be a single paragraph, there are too many blank spaces before and after Section 2 Related Work, and the spacing of equations is large. With ample space in the appendix, the full text does not fill up the nine pages. Improvements are needed in basic writing formatting and readability.
2. The paper has limited novelty. From a personal perspective, the innovation lies in using a teacher model to predict probabilities. The rest of the paper mainly demonstrates the significant impact of tail classes. From the perspective of innovative design, it is not convincing. It is also unclear why the teacher model can provide accurate estimates of $p_x$.
3. There is limited description regarding fair comparisons. After introducing the teacher model, the training process and time will be affected. The main part should devote more space to introducing the teacher model, analyzing its performance, and comparing the training time with other models.

**Questions:**

1. Can the teacher model be adapted to the current task? How does its performance compare? What are the differences between directly using a better model for distillation and utilizing the results of other models as auxiliary information in this paper?
2. Refer to the weaknesses mentioned.

---

> ### Author Response · Authors · 2023-11-19
>
> We thank the reviewer for their valuable feedback.
>
> **Can the teacher model be adapted to the current task?  How does its performance compare?**
>
> The teacher model is adapted to the current task, the Siamese model is trained on the same dataset on which the classifier-based methods were trained. The teacher model (Siamese network) is much better than the student model (classifier) on the tail (indicated by superior PSP Coverage), while it lags behind significantly in Precision. Using the superior performance of the teacher on the tail, LEVER improves the OvA classifier’s tail performance. We have updated the draft to include a comparison between teacher and student in Table 4 in the main paper.
>
> **What are the differences between directly using a better model for distillation and utilizing the results of other models as auxiliary information in this paper?**
>
> We request the reviewer to clarify the question. From what we understand, the reviewer is asking the effect of using a better model for distillation.
> If so, it is the case that a better teacher model results in better performance of classifiers when combined with LEVER. Table 13 in the supplementary shows the effect of using different teachers. We experiment with 3 teachers: Astec, MiniLM 3-layer, and 6-layer DistilBERT.However, it is important to note that even our best teacher, NGAME 6-layer DistilBert, performs much inferior to classifier-based methods on precision.
>
> **Readability of paper not strong and uncomfortable formatting. Full text not filling 9 pages. More discussion around train time and teacher model performance.**
>
> We apologize for the concerns around formatting. We have made the following changes:
> (i): Abstract has been made concise,
> (ii): Empty space around Section 2 related work and  equations has been cut down,
> (iii): We have devoted the additional space obtained to discuss the train time overhead of using LEVER (Table 3) and Comparison with the teacher model (Table 4)
>
> **Limited novelty: the innovation lies in using a teacher model to predict probabilities. From a personal perspective, the innovation lies in using a teacher model to predict probabilities From the perspective of innovative design, it is not convincing.**
>
> While the approach might be simple, we believe the concept of label variance we propose is novel. Additionally, we would like to highlight the difference between our setup and conventional distillation, which usually relies on a teacher model that is uniformly better than the student. In our work, we show that a teacher model that excels on a subpopulation (tail-labels) can also be leveraged to improve the performance of OvA classifiers for XMC tasks. Further, the simplicity of the approach should rather be viewed as a strength as it allows for easy and effective combinations with multiple SOTA classifier-based methods.
>
> **Why does the Siamese model provide accurate estimates of $p_x$?**
>
> Recent studies have shown that Siamese Networks, when used as input encoders, exhibit impressive performance on tail labels (Dahiya et al., 2021a; 2022; Jain et al., 2023). This success can be attributed to the ability of Siamese encoders to learn correlations by utilizing label-side features. Further, we consume only high confidence estimates of $p_x$ from Siamese teacher models: “For each label l, the τ nearest points from the pool are selected and added as ground truth.”
>
> ---
>
> We hope this rebuttal clarifies the concerns you had about the paper. We would be happy to discuss any further questions about the work, and would really appreciate an appropriate increase in the score if the reviewer’s concerns are adequately addressed to facilitate the acceptance of the paper.

---

> > ### Comment · Reviewer_UBHy · 2023-11-20
> >
> > Thank you for your response, but I still choose to keep my score.

---

### Official Review · Reviewer_Kx6v · 2023-10-31

**Soundness:** 2 fair
**Presentation:** 3 good
**Contribution:** 3 good
**Rating:** 8
**Confidence:** 5

**Summary:**

This paper aims to improve the tail labels performance of the extreme multi-label classification (XMC) problems. The paper propose LEVER, a knowledge distillation framework, that learns student OVA classifiers with binary relevance true labels and soft-labels generated by teacher bi-encoders. The author claims that bi-encoders have better performance on tail labels, and hence using their soft labels as additional signals to learn the student OVA classifiers can help reduce the variance for tail labels. Empirically, LEVER demonstrated consistent improvement upon competitive OVA classifiers on a wide-range of XMC datasets.

**Strengths:**

1. The paper writing is easy to follow
2. Empirical results are strong

**Weaknesses:**

1. Its questionable that using "pairwise" logistic loss (Eq.10) lead to calibrated scores
2. Some experiment settings are not clear from the main text. See detailed questions below.
3. Missing proper evaluation metrics (i.e., Macro-average Precision/Recall/F1) for tail-labels performance

**Questions:**

## 1. Calibrated Scores
Its well known that pairwise ranking losses (e.g., Equation (10) of this submission) is shift-invariant hence may not produce calibrated scores, as pointed out by [1]. Why not considering point-wise loss functions or hybrid objectives [1] (listwise Softmax + pointwise BCE) to produce more calibrated scores?

## 2. Experiment Settings and Results
(1) What's the model size for each method in Table 1 and Table 2? The model size should include every component needed to do inference. For example, does ELIAS and ELIAS+LEVER have the same model size, as LEVER is just in-place modifying the OVA classifiers of LEVER? If not, I am concern about the performance gain of LEVER is due to additional model capacity.

(2) In Table 1, ELIAS+LEVER achieved >20% absolute gain on LF-AOL-270K in P@3 and P@5. If not a typo, any insight why such significant gain? Similar questions to ELIAS+LEVER on LF-Wiki-1M, PSP metrics.

(3) On page 9, the author claims the training time of LEVER only increased by at most 2x. Does that include (a) training time of teacher bi-encoders (b) prediction time of teacher bi-encoder to generate soft-labels (c) training time of student OVA classifiers which are trained by not only sparse ground truth labels but also dense soft-labels? If so, what's the detailed breakdown in terms of those three components?

(4) Suppose the ground truth label matrix is a sparse $N \times L$ matrix. How dense are the soft-labels generated by LEVER? Does the performance of LEVER vary when using different top-$k$ soft-labels per input?

## 3. Macro-averaged Evaluation Metrics
To properly measure the performance of long-tailed labels, text classification community often consider Macro-average Precision/Recall/F1 metrics [2,3,4]. The author should also report Macro-averaged metrics to further validate the major claim of LEVER, which is the the performance gain on tailed-labels.


## Reference

[1] Yan et al. Scale Calibration of Deep Ranking Models. KDD 2022.

[2] Zhang et al. Long-tailed Extreme Multi-label Text Classification by the Retrieval of Generated Pseudo Label Descriptions. EACL 2023.

[3] Yang et al. A re-examination of text categorization methods. SIGR 1999.

[4] Lewis et al. RCV1: A New Benchmark Collection for Text Categorization Research. JMLR 2004.

---

> ### Author Response · Authors · 2023-11-19
> **Part-1 of Response**
>
> We thank the reviewer for their valuable feedback. We request the reviewer to clarify the ethical concerns with our submission as we see our paper has been flagged for ethics review.
>
> Below we clarify some of the concerns.
>
> **Calibration concerns**
>
> Thank you for bringing this paper [1] to our notice. Since our loss is shift-invariant, the scores produced by the model are not calibrated, as pointed out in [1]. If $s_x =z_l^{T}z_x$  is the score output by the Siamese model for a label $l$ (with embedding $z_l$) and document $x$ with embedding $z_x$ this would mean that $s_x$ is not calibrated, we agree. However as indicated in theorem 2, we make use of probabilities $p_x = 1/(1 + e^{-(ms_x + c)})$ where $m,c$ are hyper-parameters for training OvA classifiers. Note that this  is in line with the post-hoc calibration strategies explored in [2], where a model is learned and then a parametrized Sigmoid function is fit to learn the relevance probabilities, i.e., given an uncalibrated score $s_x$ we learn $p_x$ by fitting the sigmoid function $p_x = 1/(1 + e^{-(ms_x +c)})$ where $m, c$ are model hyper-parameters.
>
> **Why not use BCE Loss or methods proposed in [1] to achieve calibration?**
>
> We acknowledge that exploring the techniques in [1] could certainly be an interesting direction for future investigation to understand the effect of teacher models trained with different loss functions on student model performance, and we will add this as a future direction in our updated draft.
> Additionally, from our experience using BCELoss while training Bi-encoders leads to training instability. Hence we resorted to using pair-wise losses and posthoc calibration techniques
>
> [1]: Scale Calibration of Deep Ranking Models, Le Yan et al, KDD 2022
>
> [2]: Probabilistic Outputs for SVMs and Comparisons to Regularized Likelihood Methods, Platt 2000
>
> **Experimental settings and results**
>
> **Model Size**
>
> Please note that the inference pipeline is unchanged with the addition of LEVER (both in terms of model capacity and inference time). LEVER only affects the training strategy by the addition of soft labels. So, the model size is the same for the base classifier and its LEVER counterpart.
> The below table shows the model size (in Millions of parameters) for the different models and datasets. Note the smaller model size in ELIAS for LF-AmazonTitles-1.3M is due to using 512 dimension classifiers, (as 768D led to OOM issues).
>
> ||Renee|ELIAS|CascadeXML|
> |-|-|-|-|
> |LFAT-13K|167.03M|168.73M|268.94M|
> |LFA-131K|167.03M|168.73M|268.94M|
> |LF-AOL-270K|275.26M|277.14M|377.95M|
> |LF-WikiSeeAlso-320K|312.12M|315.59M|408.33M|
> |LF-Wikipedia-500K|451.90M|457.93M|553.47M|
> |LF-WikiHierarchy-970K|815.93M|829.51M|918.86M|
> |LF-AmazonTItles-1.3M|1064.76M|746.25M|1171.11M|
>
>
>
> **Significant gain in  P@3/5 AOL in ELIAS**
>
>
> We apologize for the typo here, we accidentally entered values for ELIAS + LEVER on LFAT-1.3M here. The correct values were present in the appendix (Table 7), and now they have been added to the main paper in Table 1. as well. Here are the values for your reference
>
> |LF-AOL-270K|P@1|P@3|P@5|PSP@1|PSP@3|PSP@5|
> |-|-|-|-|-|-|-|
> |ELIAS |40.83 |22.33 |14.91 |13.29 |21.46 |25.22 |
> |ELIAS + LEVER |40.85 |22.83 |15.57 |13.68 |24.30 |30.43|
>
> **Significant gain in PSP metrics in WikiHierarchy in ELIAS**
>
> In ELIAS there is a bigger drop in head performance as compared to Renee when LEVER is applied.  This therefore translates to a much larger increase in ELIAS+LEVER’s tail performance for this dataset, note that for ELIAS the performance improvement starts from the torso labels itself. This is illustrated in Figure 8 in the supplementary which compares decile wise plot ELIAS, ELIAS + LEVER and Renee, Renne+LEVER.

---

> > ### Author Response · Authors · 2023-11-19
> > **Part-2 of Response**
> >
> > **Training time**
> >
> > **On page 9, the author claims the training time of LEVER only increased by at most 2x. Does that include (a) the training time of teacher bi-encoders (b) the prediction time of teacher bi-encoders to generate soft labels (c) the training time of student OVA classifiers which are trained by not only sparse ground truth labels but also dense soft labels? If so, what's the detailed breakdown in terms of those three components?**
> >
> > The train times reported in the paper only included component (c), i.e., the training time of the student OvA classifiers. Following are the observations
> > Including all the components (a), (b), & (c) results in an average increase of 3.1x in training time and a maximum increase of 8.9x. Note that this level of increase happens only in two cases: while training CascadeXML on LF-Amazon-131K & LF-WikiSeeAlso-320K.
> >
> > In the case of the state-of-the-art method Ren\'ee, the increase in training time is at max 1.27x times the original model training time as Ren\’ee depends on the Siamese teacher for initialization.
> >
> > CascadeXML & ELIAS do not depend on the Siamese Teacher for initialization. The below table shows the total training (a) + (b) + (c) [This table has been added to main paper - Table-3). For a breakdown of these values, please refer to Table 24,25,26 in appendix.
> >
> > |Dataset|Renée|Renée+LEVER|ELIAS|ELIAS+LEVER|Cascade|Cascade+LEVER|
> > |-|-|-|-|-|-|-|
> > |LF-AmazonTitles-131K|17.59|20.11|4.33|18.82|3.63|17.14|
> > |LF-Amazon-131K|42.77|46.51|19.44|65.62|4.60|41.31|
> > |LF-AOL-270K|136.22|138.20|60.67|171.20|42.12|152.00|
> > |LF-WikiSeeAlso-320K|86.42|95.19|25.33|111.46|12.40|88.52|
> > |LF-Wikipedia-500K|154.93|184.05|138.67|226.72|29.58|89.72|
> > |LF-WikiHierarchy-1M|31.44|40.15|24.00|61.17|9.85|35.91|
> > |LF-AmazonTitles-1.3M|154.39|186.45|40.00|158.23|70.00|202.93|
> > |AverageTimeInc.||1.14x||3.29x||4.86x|
> >
> > **Effect of density on performance**
> >
> > **How dense are the soft labels generated by LEVER? Does the performance of LEVER vary when using different top-k soft-labels per input?**
> >
> > As we increase $k$, LEVER adds more samples (documents or labels). Adding more samples results in an improved tail performance, but impacts precision due to losses on head deciles. The table below (added as Table 22 in suppl.) shows the density in terms of points per label (PPL) and $k$. Figure 7 in the suppl. Shows the decile-wise plots for these configs. Kindly note that $k$ is referred to as $\tau$ in the tables and figures.
> >
> > |$k$|PPL|P@1|P@3|P@5|PSP@1|PSP@3|PSP@5|C@1|C@3|C@5|
> > |-|-|-|-|-|--|--|--|-|-|-|
> > |LF-Wikipedia-500K|||||||||||
> > |1|18.1|84.96|66.26|51.68|37.10|50.27|55.68|22.90|50.08|61.59|
> > |20|39.5|85.14|66.90|52.35|38.39|52.33|58.04|24.72|53.45|65.43|
> > |45|64.5|85.02|66.43|52.05|42.51|54.86|60.20|29.08|58.28|70.09|
> > |**LF-WikiHierarchy-1M**|||||||||||
> > |1|43.3|95.01|93.99|92.24|19.69|27.36|33.20|6.62|11.39|14.56|
> > |4|44.8|95.19|93.90|92.07|24.79|32.74|38.29|9.08|16.12|20.02|
> > |8|47.7|94.97|93.18|90.98|26.37|34.99|40.51|10.08|18.34|22.74|
> >
> > **Macro Metrics**
> > As recommended in [3] we report MacroF1, MarcoPrecision, & MacroRecall at $k$ where $k$ values for datasets were chosen based on average labels per point for the dataset. We use $k$=3 for LF-AmazonTitles-131K, LF-Amazon-131K, LF-WikiSeeAlso-320K, LF-AOL-270K, $k$=5 for LF-Wikipedia-500K, $k$=25 for LF-AmazonTitles-1.3M and LF-WikiHierarchy-1M. We see an average gain of $4.3%$ in MacroF1, $4.1%$ in MacroPrecision, & $5.6%$ in MacroRecall. The below table summarizes the results [Added as Table 8 in suppl.]
> >
> > |**Model**|LF-AmazonTitles-131K|||LF-Amazon-131K|||LF-Wikipedia-500K|||
> > |-|-|-|-|-|-|-|-|-|-|
> > ||F1@k|P@k|R@k|F1@k|P@k|R@k|F1@k|P@k|R@k|
> > |ELIAS|24.23|22.82|31.70|28.48|26.46|37.93|27.98|27.97|35.17|
> > |ELIAS+LEVER|29.53|27.70|38.21|33.36|31.16|43.35|34.12|33.88|44.68|
> > |CascadeXML|22.23|20.80|30.36|28.89|26.74|38.83|20.04|20.13|26.75|
> > |CascadeXML+LEVER|29.12|26.91|39.04|33.29|30.76|44.13|31.40|31.61|41.70|
> > |Renee|32.19|30.36|41.55|32.55|29.95|43.88|35.94|36.78|43.76|
> > |Renee+LEVER|33.35|31.78|42.45|35.69|34.05|45.04|40.54|40.71|51.36|
> >
> > ||**LF-AmazonTitles-1.3M**|||**LF-AOL-270K**|||**LF-WikiHierarchy-1M**|||
> > |-|-|-|-|-|-|-|---|-|-|
> > ||F1@k|P@k|R@k|F1@k|P@k|R@k|F1@k|P@k|R@k|
> > |ELIAS|16.43|15.73|24.68|10.69|9.45|16.18|20.71|23.26|22.43|
> > |ELIAS+LEVER|18.85|17.92|28.58|13.97|12.87|19.41|26.99|28.75|30.53|
> > |CascadeXML|13.58|13.23|22.06|9.00|8.25|13.46|17.73|20.91|19.22|
> > |CascadeXML+LEVER|16.02|15.07|27.02|11.59|11.03|15.74|21.17|24.36|21.85|
> > |Renee|24.79|24.16|35.42|17.7|16.88|22.53|24.48|27.59|26.04|
> > |Renee+LEVER|26.72|25.73|37.04|22.38|20.4|30.18|27.89|30.67|30.23|
> >
> > [3]:Zhang et al. Long-tailed Extreme Multi-label Text Classification by the Retrieval of Generated Pseudo Label Descriptions. EACL 2023.
> >
> > We hope this rebuttal clarifies the concerns you had. We would be happy to discuss any further questions about the work, and would appreciate an appropriate increase in the score if the reviewer’s concerns are addressed to facilitate the acceptance of the paper.

---

> > > ### Comment · Reviewer_Kx6v · 2023-11-22
> > >
> > > I am satisfied with the authors response and appreciate the additional experiment results. I am happy to increase the rating.

---

> > > > ### Author Response · Authors · 2023-11-22
> > > > **Thank you!**
> > > >
> > > > We're happy to know that our response helped in addressing your concerns. Thanks again for the helpful feedback and for upgrading the score!

---

### Official Review · Reviewer_Cq1J · 2023-11-06

**Soundness:** 3 good
**Presentation:** 3 good
**Contribution:** 3 good
**Rating:** 6
**Confidence:** 4

**Summary:**

The paper introduces a new extreme multi-label classification algorithm called LEVER, that aims to improve the existing methods by proposing training with a loss function that combines loss calculated on observed hard labels and soft labels coming from the teacher model, which is assumed to have superior performance on the tail labels. The teacher model used by the authors in the empirical part of the paper is a Siamese-style neural network that leverages label features and is trained with a logistic-loss-based objective, which was found to yield calibrated estimates of the marginal probabilities of labels. The authors demonstrate the attractiveness of the proposed approach in the exhaustive empirical comparison when they used 4 popular XMLC benchmarks, introduced 2 new benchmarks, and combined the proposed approach with 3 SOTA methods. The proposed approach, in many cases, significantly improves the performance on standard precision@k, propensity-scored precision@k, and coverage@k.

**Strengths:**

1. The paper is well-written and easy to follow.
2. The proposed method is simple to apply to a wide range of XML classifiers.
3. The exhausting empirical comparison proved the attractiveness of the proposed method.
4. In addition to the main results, the authors provide a wide array of additional experiments in the appendix.
5. Two new datasets additionally strengthen the contribution of this work.

**Weaknesses:**

1. The proposed method is just a combination of loss with hard and soft labels, which is a simple idea.
2. I find the theoretical results rather simplistic, expected, and being there to serve as just justification for the applied method rather than its original motivation.
3. This part of the paper especially confirms that for me:

   > Notably, if we have precise estimates of marginal relevance, denoted by $p_x = E[y|x]$, we can replace $y$ with $p_x$, effectively reducing the variance term to 0 and thereby improving classifier generalization. This principle forms the foundation for the LEVER framework, which employs an additional teacher network to provide accurate estimates of $p_x$.

   If we have a model that provides good estimates $p_x$ then we don't need to train another one, the XMLC task is already solved! The strength of the method seems to lay in the properly selected trade-off between loss calculated on hard observed labels and soft labels coming from the teacher network that seems to be much better on tail-labels thanks to leveraging labels-side information.

4. From the appendix, I understand that $\lambda = 0.05$ (the variable that weights the hard and soft part of the loss) was used for all the experiments. Since the choice of $\lambda$ seems to be crucial for the method. I find the lack of further comments on it and experiments demonstrating its impact the biggest weakness of this paper.

My score is 6, but I believe the paper is even a bit above that.

**Questions:**

1. As I mentioned in the weakness section, I would like to see how different values of $\lambda$ impact the final performance and what the trade-off curve between head-labels and tail-labels performance looks like. Could the authors comment on that?
2. On the LF-AOL-270K dataset, LEVER combined with ELIAS yields extremely high improvement on standard precision@k, especially at @5. These seem almost unrealistic when compared with scores of other methods. Are these numbers for sure correct? If yes, do the authors have any explanation for this result?

---

> ### Author Response · Authors · 2023-11-19
>
> We thank the reviewer for their valuable feedback. Below, we provide clarifications.
>
> **The proposed method is just a combination of loss with hard and soft labels, which is a simple idea.**
> Yes, while the idea is a combination of hard and soft labels, we would like to highlight that this simplicity allows the easy and effective combination of LEVER with any OvA-based XMC approach. However, unlike standard distillation where logits for all classes from teacher are used, in our case, for each label l, only the τ nearest points from the pool are selected and added as ground truth for distillation. This is how we have adapted distillation for our multi-label scenario. Table 22 and Figure 7 in supplementary show that increasing τ improves performance on tail, while it hurts head or torso label
>
> **If we have a model that provides good estimates then we don't need to train another one; the XMLC task is already solved!**
>
> Our intention for writing “if we have precise estimates of marginal relevance.. Reducing the variance to 0” was to convey that as the teacher gets more accurate, the variance term reduces to zero, and thus the upper bound on the true risk reduces, thereby improving generalization error of the classifier. However, a perfect teacher model is not available in practice. The NGAME model we use is a proxy for the perfect teacher and note that it is more accurate than the OvA classifier only on tail labels. Leveraging the strength of this Siamese model on the tail to reduce the label variance of OvA classifiers on the tail is what LEVER aims to achieve.
>
> **From the appendix, I understand that $\lambda$ (the variable that weights the hard and soft part of the loss) was used for all the experiments..**
>
> We think the reviewer has confused the $\lambda$ for ELIAS mentioned in Section D.2 of supplementary with the $\lambda$ for LEVER. We have renamed the $\lambda$ used in Section D.2 to $\lambda_{elias}$. The $\lambda$ used in LEVER is 0.5 (equal weight to both hard & soft labels).
>
> **I would like to see how different values of $\lambda$ impact the final performance and what the trade-off curve between head-labels and tail-labels performance looks like**
>
> In table below, we show the effect of varying $\lambda$ when LEVER is combined with Renee on 3 datasets: LF-AmazonTitles-131K, LF-Wikipedia-500K, & LF-AOL-270K. Note that $\lambda$=0.33 weighs the soft labels with double the weight as compared to hard labels.
> Our observations are:
> The equal weightage (0.5) gives the best performance.
> Increasing $\lambda$ weighs the hard labels more and this results in performance that gets closer to the base classifier, i.e., PSP worsens, and Precision remains flat or drops slightly.
> Additionally, decreasing $\lambda$ helps only up to a certain point, i.e., $\lambda$=0.5; this is because our teacher is not perfect we need to strike a balance between hard and soft labels.
>
> |LF-AmazonTitles-131K||||||||||
> |-|-|-|-|-|-|-|-|-|-|
> |$\lambda$|P@1|P@3|P@5|PSP@1|PSP@3|PSP@5|C@1|C@3|C@5|
> |0.33|46.15|30.76|21.87|39.70|45.33|50.13|32.56|54.48|61.24|
> |0.50|46.44|30.83|21.92|39.70|45.44|50.31|32.82|55.11|61.94|
> |0.66|46.57|30.87|21.93|39.59|45.46|50.36|32.31|54.49|61.48|
> |0.80|46.58|30.88|21.92|39.37|45.39|50.33|32.06|54.38|61.42|
>
> |LF-Wikipedia-500K||||||||||
> |-|-|-|-|-|-|-|-|-|-|
> |$\lambda$|P@1|P@3|P@5|PSP@1|PSP@3|PSP@5|C@1|C@3|C@5|
> |0.33|84.66|65.94|51.54|40.51|53.40|58.75|26.88|56.01|67.94|
> |0.50|85.02|66.42|52.05|42.50|54.86|60.20|29.46|58.53|70.29|
> |0.66|84.96|66.51|52.18|39.34|53.53|59.46|25.66|55.78|68.33|
> |0.80|84.84|66.42|52.14|38.66|53.09|59.16|24.96|55.08|67.79|
>
> |AOL-270K||||||||||
> |-|-|-|-|-|-|-|-|-|-|
> |$\lambda$|P@1|P@3|P@5|PSP@1|PSP@3|PSP@5|C@1|C@3|C@5|
> |0.33|41.14|24.00|16.50|17.24|32.31|40.02|14.14|36.67|45.84|
> |0.50|41.70|24.78|17.07|20.38|37.07|45.13|17.43|42.54|52.01|
> |0.66|41.44|24.41|16.76|16.76|32.69|40.47|14.16|37.38|46.54|
> |0.80|41.17|24.04|16.49|15.59|30.30|37.64|13.16|34.62|43.25|
>
> This table has been added as Table 16 in the supplementary material. For decile-wise plots showing head-tail variation, please refer to Fig. 4, 5, 6 in the supplementary material
>
> **On the LF-AOL-270K dataset, LEVER combined with ELIAS yields extremely high improvement on standard precision@k..**
>
> Thanks for pointing this out. This is a typo; the number from the LF-AmazonTitles-1.3M ELIAS+LEVER row was mistakenly copied to the ELIAS+LEVER row in LF-AOL-270K. The supplement however has the correct values for LF-AOL-270K without this typo. We fixed this in the main table of updated draft.
> Here are the values for your reference
> |LF-AOL-270K|P@1|P@3|P@5|PSP@1|PSP@3|PSP@5|
> |-|-|-|-|-|-|-|
> |ELIAS |40.83 |22.33 |14.91 |13.29 |21.46 |25.22 |
> |ELIAS + LEVER |40.85 |22.83 |15.57 |13.68 |24.30 |30.43|
>
> ---
> We hope this rebuttal clarifies your concerns. We would be happy to discuss any further questions, and would appreciate an appropriate increase in the score if the reviewer’s concerns are addressed to facilitate the acceptance of the paper.

---

> > ### Author Response · Authors · 2023-11-22
> >
> > Since the author-reviewer discussion period ends soon, we request the reviewer check our responses. We would appreciate an appropriate increase in the score if the reviewer’s concerns are adequately addressed to facilitate the acceptance of the paper.

---

> > > ### Comment · Reviewer_Cq1J · 2023-11-23
> > > **Re: Official Comment by Authors**
> > >
> > > I thank the authors for additional clarifications. I also checked the other reviews and responses. I'm generally satisfied with the additional experiments conducted by the authors. The experimental setup and a large number of experiments are big strengths of the paper and the authors' rebuttal only improves in this area. However, the discussion does not change my opinion on the limited novelty of this work and the weak and unoriginal theoretical justification for the idea. If that it would be possible, at this point, I would give this paper a 7, but right now, I'm keeping my score as it is.
> > >
> > > Additional comments:
> > > The $\tau$ parameters seem to have a significant impact on model performance. I believe it should be already discussed and motivated in Section 3 of the paper.
> > >
> > > NIT:
> > > I found a broken reference in the appendix in the added section about the impact of $\lambda$:
> > > > Effect of varying λ: The hyperparameter λ controls the importance between the two loss terms. Table 16 shows the effect of varying λ, and Figures ?? show their corresponding decile-wise plots

---

> > > > ### Author Response · Authors · 2023-11-23
> > > >
> > > > We thank the reviewer for their valuable feedback! About novelty, we would like to highlight our contribution in terms of putting forward the concept of label variance. To the best of our knowledge, this concept has not been explored in prior XML literature. We would also like to clarify that Menon et al. work [1] develops a relation between variance in loss and the model's generalization performance (Proposition 2). Our work builds upon this to create a link between label variance and generalization performance (Theorem 1). As pointed out by reviewer Uo9h, we will work on polishing our presentation of the theoretical results in our updated draft.
> > > >
> > > > We will also discuss the effect of $\tau$, currently discussed in Section 5, Pg 9 in Section 3, and address the broken reference issue you have highlighted in our updated draft
> > > >
> > > > [1]: A Statistical Perspective on Distillation. Menon et al. 2021

---

### Official Review · Reviewer_Uo9h · 2023-11-09

**Soundness:** 2 fair
**Presentation:** 2 fair
**Contribution:** 2 fair
**Rating:** 6
**Confidence:** 3

**Summary:**

The paper presents a methodology for improving performance on tail-labels. The main observation behind the proposed approach is that there is a significant variance in the label distribution due to the finite and even scarce number of data points for tail-labels in the extreme classification. The paper builds a connection with an existing work (Menon et al. 2021), on which it heavily relies, for most of the motivation, to propose a variance reduction strategy (Theorem 1 in the paper & discussion thereafter) and hence improvement in the generalization error.

The focus of the approach (generalization analysis) is on the last/classification layer of the network. The above generalization analysis leads to an augmented objective which, in addition to standard hard labels, also consists of a loss term consisting of soft labels from a teacher model, for which recent frameworks based on Siamese training are exploited. The proposed approach is tested on a range of datasets from the extreme classification repository, and it is shown that the proposed approach leads to significant improvements in the prediction performance in terms of P@k and PSP@k metrics. The augmentation strategy is also compared to existing methods for data augmentation in extreme classification to further demonstrate its general applicability.

**Strengths:**

1. The paper attempts to address a key shortcoming of existing methods in extreme classification i.e. performance of sota methods on tail-labels, which despite its importance, doesn’t get much focus of research.

2. The experimental results of the paper are quite impressive, and significant gains on a range of datasets, and methods are shown, thereby demonstrating its general applicability. Overall, the experimental setup is quite detailed.

3. The paper also contributes two additional datasets (LF-AOL-270K, and LF-WikiHierarchy-1M), which may be helpful for the community.

**Weaknesses:**

1. The main shortcoming of the paper is the lack of novelty in the main idea and the approach :

1a) In terms of the content i.e. the main theoretical results such as Theorem 1, its proof, the idea to reduce variance for improving generalization, using soft-labels in teacher-student setup has already been explored in the existing work of Menon et al 2021. It seems that the paper attempts to build a loose bridge between this earlier theoretical work to build a weighting scheme and soft-labels (equation 9).

1b) The paper claims that the variance issue for tail-labels has not been explored so far. However, this seems not quite true, as a similar concern has been raised in the earlier work Babbar and Scholkopf 2019, where there was a similar argument that due to the lack of data in tail-labels, for a given class label, there is high variance between the input features between two samples.

2) There also seems to be a lack of consistency in terms of using the symbols and notation in many places. For instance, X is used in Section 3.1 to denote a (random?) instance in the input space $\mathcal{X}$. However, the same is used to limit the norm of x in the same section. For equation (1), should the LHS also not be conditioned on x i.e. V[y|x]. It is not clear why there are two different symbols for the sub-script (small-case) x. Why the equation (6) is defined in terms of a trained classifier w*, while the standard generalization results are stated in terms of all possible classifiers in the hypothesis class. The theorem statement needs to be on a better formal footing as done in the original paper Menon et al. 2021, the current version is incomplete and unclear.

3. In terms of properly citing related work, the paper falls somewhat short. For instance, the paper introduces the need to use calibrated losses without a proper justification to use calibrated losses or a reference thereof. It does not seem to be in the sense of the word used in the last sentence of page 1. Another concern is the lack of setting the right context for the cited papers.  As mentioned above, the high variance problem for tail-labels has also been explored. As another instance, the work of Schultheis et al. 2022 is cited but the correct context is in the related work section when discussing Missing and Tail-labels in Section 2.2, and not only later down in discussing coverage as a metric. Even though the paper has been cited, it is not in the right context. In terms of the main classifier, the focus of the paper is on one-vs-rest approaches, while DiSMEC which initiated this approach in extreme settings isn’t referred.

**Questions:**

As mentioned above in the weaknesses section

---

> ### Author Response · Authors · 2023-11-19
>
> We thank the reviewer for their valuable feedback. Below, we provide clarifications:
>
> >  Concerns around novelty: The paper claims that the variance issue for tail labels has not been explored so far. However, this seems not quite true, as a similar concern has been raised in the earlier work by Babbar and Scholkopf 2019
>
> [1] discusses the issue of variance from the point of view of a lack of commonality between features of training instances and those between training and test instances. However, the definition of variance we use is different, and we highlight it with an example below:
>
> Consider a dataset where data points are user queries and labels are advertisements shown in response to user queries.
> Now, if we consider a label with the text “insurance,” there could be a wide variety of user queries (or documents), such as “hospital bills” and “car repair,” that are relevant to this label. This lack of commonality between documents associated with a single label has been discussed in [1].
>
> In recommendation tasks, dataset rely on click signals to build the ground truth, i.e., if the user clicks on an advertisement for a particular query, then the query-advertisement pair is considered relevant (1 in ground truth). However, feedback in the form of user clicks is subject to variability as a user’s interests can fluctuate over time; for example, for the query “hospital bills,” there may or may not be a click on the “insurance” ad. Approximating the relevance using click signals that are binary leads to inaccuracies in the dataset, which we quantify using the term label variance.
>
> In regards to the novelty of our work, we would like to highlight a few points:
>
> 1.	We propose the concept of label variance, which, to the best of our knowledge, has not been explored in the domain of extreme classification.
>
> 2.	While Menon et al. discuss the teacher-student setup, their teacher is better than the student overall (as indicated by superior Precision values in Table 1). While such a strong teacher might not always be available, we demonstrate the potential of improving the performance of OvA classifiers by proposing a Siamese teacher that excels only  on tail labels. Note that the Siamese teacher is inferior in precision compared to the OvA student model. To the best of our knowledge, we are the first to propose distilling from a teacher that excels only on a subset of the dataset population (tail labels) for XMC tasks.
>
> 3.	While the idea is simple, we believe the method's simplicity is a strength as it allows for an easy and effective combination with state-of-the-art one-versus-all methods, as highlighted in our results.
>
> [1]: “Data scarcity, robustness, and extreme multi-label classification” (Babbar and Scholkopf 2019)
>
> > Inconsistent notation in Section 3.1, and better presentation of improving presentation of Theorem 1.
>
> Thanks for highlighting this. We will work on addressing the issues in our final draft.
>
> > Citing related works
>
> **Justification for the use of calibrated losses**
>
> In Section 3.3, we discuss the need to use calibrated losses. In summary, Siamese networks trained using triplet loss have shown to do well on tail labels. However, the scores output by these models are not well calibrated, i.e., they do not have a probabilistic interpretation). For training OvA classifiers with soft targets, we need calibrated scores, and Theorem 2 in our paper shows how we can obtain calibrated scores from the Siamese teacher.  Note that this is in line with the posthoc calibration strategy mentioned in [3], where a model is learned. Then a parametrized Sigmoid function is fit to learn the relevance probabilities, i.e., given an uncalibrated score $s_x$ we get $p_x$ by fitting the sigmoid function $p_x = 1/(1 + e^{-(ms_x +c)})$ where $m, c$ are model hyper-parameters.
> We will clarify the definition of calibration used in the current context and add a citation to [2].
>
> **Meaning of calibration used at the end of Pg1?**
>
> At the end of Pg 1. We discuss the strategies that rebalance the loss functions to tackle the problem of extreme class imbalance in XMC datasets. We will clarify this in the final version of the paper.
>
> **Citing work of Schultheis et al. 2022 in the correct context and missing citation of DiSMEC**
>
> In our revised draft, we have cited the work of Schultheis et al. 2022 in Section 2.2 while discussing “Bias due to Missing Lables,” and  DiSMEC has now been cited in the introduction while discussing about OvA classifiers in the revised draft.
>
> We hope this rebuttal clarifies the concerns you had about the paper. We would be happy to discuss any further questions about the work. We would really appreciate an increase in the score if the reviewer’s concerns are adequately addressed to facilitate the acceptance of the paper.
>
> [2]: Scale Calibration of Deep Ranking Models, Le Yan et al, KDD 2022
> [3] Probabilistic Outputs for SVMs and Comparisons to Regularized Likelihood Methods, Platt 2000

---

> > ### Author Response · Authors · 2023-11-22
> >
> > Since the author-reviewer discussion period ends soon, we request the reviewer check our responses. We would appreciate an appropriate increase in the score if the reviewer’s concerns are adequately addressed to facilitate the acceptance of the paper.

---

> > ### Comment · Reviewer_Uo9h · 2023-11-23
> > **post author rebuttal**
> >
> > Thanks to the authors for the response. While this does help somewhat in improving my score, there are still some concerns/clarifications required :
> > 1. Variance issues : In my opinion, both the earlier work [1] and this one focus on different aspects of the same issue i.e. data scarcity leading to shift in the observed empirical distribution between different sub-samples. In [1] the change in the empirical distribution (between train and test) of the inputs x_i for a given label, and in this case, possible change in the observed label y_i for an instance x_i over time or different annotators. In the second scenario, however, it seems like a missing label problem, even though the authors mention that this does not include any systematic bias, and hence they claim that it is not a missing labels problem, to which I am not sure if I entirely agree. Despite that PSP metrics are used for evaluation, which were designed for missing label problem in the first place, and the usage for measuring the tail-label performance as a metric is rather secondary (as per Schultheis et al 2022). Apparently, we seem to be going in circles, and this requires some clarification/discussion in the paper.
> >
> > 2. The overlap with Menon et al. 2021 remains a concern, and the paper in its current form lacks theoretical soundness in comparison to that paper. On the positive side, it has strong empirical performance and addresses a key problem in XMLC literature.

---

> > > ### Author Response · Authors · 2023-11-23
> > >
> > > We thank the reviewer for appreciating the strong empirical performance of our method.
> > >
> > > We will revise our draft to (i) clarify the difference between missing labels and label variance and (ii) Polish our presentation of theoretical results.

---

### Comment · Area_Chair_Cace · 2023-11-22
**Author-Reviewer Discussion ends soon**

Dear Reviewers and Authors,

The discussion phase ends soon. Please check all the comments, questions, and responses and react appropriately.

Thank you!

Best, AC for Paper #9297

---

### Author Response · Authors · 2023-11-23
**Rebuttal Summary**

We thank the reviewers for their efforts in reviewing our work. Below, we summarize the rebuttal.

-  As suggested by [Reviewer Kx6v](https://openreview.net/forum?id=6ARlSgun7J&noteId=jns9oikZjC), we have added a detailed breakdown of the training time overhead by adding LEVER (Table 24,25, 26 in the suppl.), clarified the model size and examined the effect of using different $\tau$ ([this response](https://openreview.net/forum?id=6ARlSgun7J&noteId=RdASL7hszP)) in Table 22 in the suppl. We are glad that all of their concerns have been resolved.

-  As suggested by [Reviewer Cq1J](https://openreview.net/forum?id=6ARlSgun7J&noteId=O6ZTIrtlbZ), we have discussed the effect of varying the $\lambda$ parameter in our loss function in [this response](https://openreview.net/forum?id=6ARlSgun7J&noteId=cALwRAOyWZ). We will discuss the effect of varying $\tau$, currently discussed in Section 5,  in Section 3 in the revised draft. We appreciate the reviewer's positive outlook on our experimental rigor.

- Lastly, as suggested by [Reviewer Uo9h](https://openreview.net/forum?id=6ARlSgun7J&noteId=bPwOn1D8KL), we have clarified the differences between our work and [1] in [this response](https://openreview.net/forum?id=6ARlSgun7J&noteId=9HrmB4lqcE). We will include this in our final draft and further polish our presentation of theoretical results in the final version, as this will require re-writing to adhere to the page limits at the moment.

[1]: "Data scarcity, robustness, and extreme multi-label classification" (Babbar and Scholkopf 2019)

---

### Meta-Review · Area_Chair_Cace · 2023-12-15

**Metareview:**

The paper concerns the problem of tail-label performance in extreme multi-label classification. As variance on tail-labels can be high, the authors introduce an architecture with an additional Siamese teacher model. The introduced solution is justified by a theoretical analysis and very promising empirical results.

The weak point of the paper raised by the reviewers is that the the introduced algorithm is based rather on a simple idea with limited novelty. Additionally, the theoretical analysis is flawed as it heavily relies on previous results without a clear extension or elaboration on them. Nevertheless, the new algorithm achieves state-of-the-art results for extreme multi-label classification tasks, motivating new research in this domain.

The authors are encouraged to improve the theoretical part in their final version of the paper.

**Justification For Why Not Higher Score:**

The paper has several flaws raised by the reviewers such as limited novelty of the algorithmic solution or limited theoretical analysis heavily relying on previous results.

**Justification For Why Not Lower Score:**

The new algorithm achieves state-of-the-art results.

---

### Decision · Program_Chairs · 2024-01-16

Accept (poster)